# The emergence of visual category representations in infants' brains

Xiaoqian Yan[1,2,3]*, Sarah Shi Tung[1‡], Bella Fascendini[1‡], Yulan Diana Chen[1,2], Anthony M Norcia[1,2†], Kalanit Grill-Spector[1,2,4†]

[1]Department of Psychology, Stanford University, Stanford, United States; [2]Wu Tsai Neurosciences Institute, Stanford University, Stanford, United States; [3]Institute of Science and Technology for Brain-Inspired Intelligence, Fudan University, Shanghai, China; [4]Neurosciences Program, Stanford University, Stanford, United States

## eLife Assessment

This **valuable** study investigates the development of high-level visual responses in infants, finding that neural responses specific to faces are present by 4-6 months but not earlier. The study is methodologically **convincing**, using state-of-the-art experimental design and analysis approaches. The findings would be of broad interest to the cognitive neuroscience and developmental psychology research communities.

**\*For correspondence:**
xqyan@fudan.edu.cn

[†]These authors contributed equally to this work
[‡]These authors also contributed equally to this work

**Competing interest:** The authors declare that no competing interests exist.

**Abstract** Organizing the continuous stream of visual input into categories like places or faces is important for everyday function and social interactions. However, it is unknown when neural representations of these and other visual categories emerge. Here, we used steady-state evoked potential electroencephalography to measure cortical responses in infants at 3–4 months, 4–6 months, 6–8 months, and 12–15 months, when they viewed controlled, gray-level images of faces, limbs, corridors, characters, and cars. We found that distinct responses to these categories emerge at different ages. Reliable brain responses to faces emerge first, at 4–6 months, followed by limbs and places around 6–8 months. Between 6 and 15 months response patterns become more distinct, such that a classifier can decode what an infant is looking at from their brain responses. These findings have important implications for assessing typical and atypical cortical development as they not only suggest that category representations are learned, but also that representations of categories that may have innate substrates emerge at different times during infancy.

## Introduction

Visual categorization is important for everyday activities and is amazingly rapid: adults categorize the visual input in about one-tenth of a second (***Thorpe et al., 1996***; ***Grill-Spector and Kanwisher, 2005***). In adults and school-age children, this key behavior is supported by both clustered and distributed responses to visual categories in high-level visual cortex in ventral temporal and lateral occipitotemporal cortex (VTC and LOTC, respectively) (***Grill-Spector and Weiner, 2014***; ***Bugatus et al., 2017***). A visual category consists of items that share common visual features and configurations (***Grill-Spector and Kanwisher, 2005***; ***Nordt et al., 2021***; ***Margalit et al., 2020***; ***Gomez et al., 2017***; ***Stigliani et al., 2015***); e.g., corridors share features of floors, walls, and ceilings, with a typical spatial relationship. Clustered regions in VTC and LOTC (***Bugatus et al., 2017***; ***Haxby et al., 2001***; ***Nordt et al., 2023***) respond more strongly to items of ecologically relevant categories (faces, bodies, places, words) than other stimuli (***Nordt et al., 2021***; ***Kanwisher et al., 1997***; ***Epstein and Kanwisher, 1998***; ***Downing et al., 2001***; ***Golarai et al., 2007***; ***Scherf et al., 2007***; ***Dehaene-Lambertz et al., 2018***) and

distributed neural responses across VTC and LOTC (*Bugatus et al., 2017*; *Haxby et al., 2001*; *Nordt et al., 2023*) are reliable across items of a category but distinct across items of different categories. However, it is unknown when these visual category representations emerge in infants' brains.

Behaviorally, infants can perform some level of visual categorization within the first year of life. Measurements of infants' looking preferences and looking times suggest that visual saliency impacts young infants' viewing patterns (*Spriet et al., 2022*): between 4 and 10 months of age, infants can behaviorally distinguish between faces and objects (*Spriet et al., 2022*; *Mondloch et al., 1999*) and between different animals like cats and dogs (*Quinn et al., 1993*; *Younger and Fearing, 1999*). Later on, between 10 and 19 months, infants behaviorally distinguish broader-level animate vs. inanimate categories (*Spriet et al., 2022*). Neurally, electroencephalographic (EEG) studies have found stronger responses to images of faces vs. objects or textures in 4- to 12-month-olds (*Conte et al., 2020*; *Farzin et al., 2012*; *de Heering and Rossion, 2015*) and that stimulus category can be decoded from distributed responses slightly but significantly above chance in 6- to 15-month-olds (*Bayet et al., 2020*; *Xie et al., 2022*). Functional magnetic resonance imaging (fMRI) studies have found stronger responses to videos of faces (*Deen et al., 2017*; *Kosakowski et al., 2022*), bodies (*Kosakowski et al., 2022*), and places (*Deen et al., 2017*; *Kosakowski et al., 2022*) vs. objects in clustered regions in VTC and LOTC of 2- to 10-month-olds. However, because prior studies used different types of stimuli and age ranges, it is unknown when representations to various categories emerge during the first year of life. To address this key open question, we examined when neural representations to different visual categories emerge during infancy using EEG in infants of four age groups spanning 3–15 months of age.

We considered two main hypotheses regarding the developmental trajectories of category representations. One possibility is that representations to multiple categories emerge together because infants need to organize the barrage of visual input to understand what they see. Supporting this hypothesis are findings of (i) selective responses to faces, places, and body parts in VTC and LOTC of 2- to 10-month-olds (*Kosakowski et al., 2022*), and (ii) above chance classification of distributed EEG responses to toys, bodies, faces, houses in 6- to 8-month-olds (*Xie et al., 2022*) as well as animals and body parts in 12- to 15-month-olds (*Bayet et al., 2020*).

Another possibility is that representations of different categories may emerge at different times during infancy. This may be due to two reasons. First, representations of ecologically relevant categories like faces, body parts, and places may be innate because of their evolutionary importance (*Kanwisher, 2010*; *Sugita, 2009*; *Mahon and Caramazza, 2011*; *Bi et al., 2016*), whereas representations for other categories may develop later only with learning (*Nordt et al., 2021*; *Nordt et al., 2023*; *Behrmann and Plaut, 2015*). Supporting this hypothesis are findings that newborns and young infants tend to orient to faces (*Pascalis et al., 1995*) and face-like stimuli (*Johnson et al., 1991*), as well as have cortical responses to face-like stimuli (*Buiatti et al., 2019*), but word representations only emerge in childhood with the onset of reading instruction (*Nordt et al., 2021*; *Nordt et al., 2023*; *Dehaene et al., 2010*). Second, even if visual experience is necessary for the development category representations (including faces; *Arcaro et al., 2019*; *Scott and Arcaro, 2023*; *Livingstone et al., 2017*; *Arcaro et al., 2017*), categories that are seen more frequently earlier in infancy may develop before others. Measurements using head-mounted cameras suggest that infants' visual diet (composition of visual input) varies across categories and age: The visual diet of 0- to 3-month-olds contains ~25% faces and <10% hands, that of 12- to 15-month-olds contains ~20% faces and ~20% hands (*Fausey et al., 2016*; *Jayaraman et al., 2017*), and that of 24-month-olds contains ~10% faces, and ~25% hands. Thus, looking behavior in infants predicts that representations of faces may emerge before that of limbs.

## Results

45 infants from four age groups: 3–4 months (n=17, 7 females), 4–6 months (n=14, 7 females), 6–8 months (n=15, 6 females), and 12–15 months (n=15, 4 females) participated in EEG experiments. Twelve participants were part of an ongoing longitudinal study and came for several sessions spanning at least 3 months apart. Infants viewed gray-scale images from five visual categories present in infants' environments (faces, limbs, corridors, characters, and cars) while EEG was recorded. Different from prior infant studies (*Conte et al., 2020*; *de Heering and Rossion, 2015*; *Bayet et al., 2020*; *Xie et al., 2022*; *Deen et al., 2017*; *Kosakowski et al., 2022*), we used images that have been widely used in fMRI studies (*Nordt et al., 2021*; *Gomez et al., 2017*; *Allen et al., 2022*; *Lerma-Usabiaga*

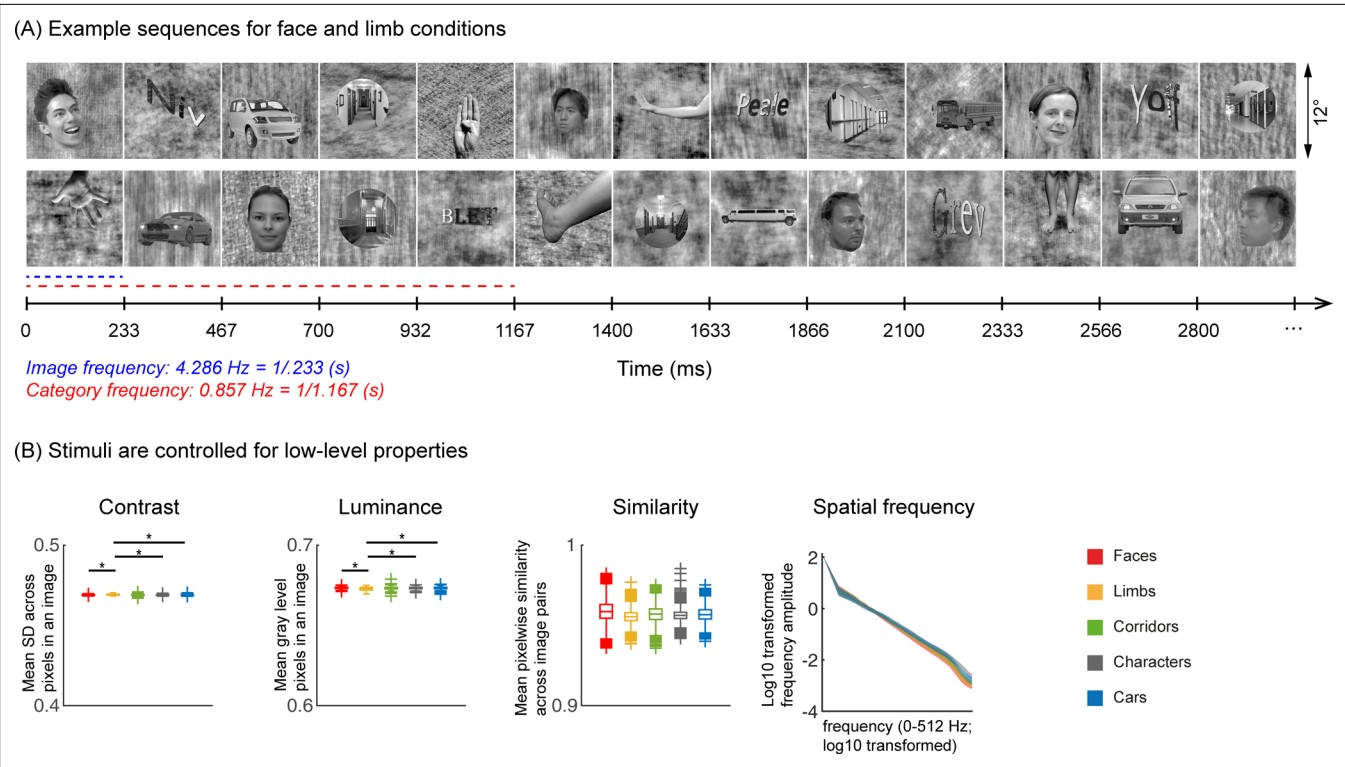

**Figure 1.** Experimental design and stimuli analysis. (**A**) Example segments of presentation sequences in which faces (top panel) and limbs (bottom panel) were the target category. Images spanning 12° containing gray-level images of items from different categories on a phase-scrambled background appeared for 233 ms (frequency: 4.286 Hz). A different exemplar from a single category appeared every fifth image (frequency: 0.857 Hz). Between the target category images, randomly drawn images from the other four categories were presented. Sequences consisted of 20% images from each of the five categories and no images were repeated. Each category condition lasted for 14 s and contained 12 such cycles. Participants viewed in random order 5 category conditions: faces, limbs, corridors, characters, and cars forming a 70 s presentation sequence. (**B**) Images were controlled for several low-level properties using the SHINE toolbox as explained in *Stigliani et al., 2015*. Metrics are colored by category (see legend). *Contrast*: mean standard deviation of gray-level values in each image, averaged across 144 images of a category. *Luminance*: mean gray-level of each image, averaged across 144 images of a category. *Similarity:* mean pixel wise similarity between all pairs of images in a category. For all three metrics, boxplots indicate median, 25%, 75% percentiles, range, and outliers. Significant differences between categories are indicated by asterisks, for contrast and luminance (nonparametric permutation t-test p<0.05, Bonferroni corrected); for image similarity, all categories are significantly different than others (nonparametric permutation testing, p<0.05, Bonferroni corrected, except for corridors vs. cars, p=0.24). *Spatial frequency: Solid lines:* distribution of spectral amplitude in each frequency averaged across 144 images in each category. *Shaded area:* standard deviation. Spatial frequency distributions are similar across categories.

*et al., 2018*; *Jagadeesh and Gardner, 2022*) and are largely controlled for low-level properties such as luminance, contrast, similarity, and spatial frequency (*Figure 1B* and *Appendix 1—table 3*). We use a steady-state visual evoked potential (*de Heering and Rossion, 2015*; *Farzin et al., 2012*; *Heinrich et al., 2009*; *Liu-Shuang et al., 2014*) (SSVEP) paradigm: In each 70 s sequence, images from five categories were shown every 0.233 s; one of the categories was the target, so different images from that category appeared every 1.167 s, and the rest of the images were drawn from the other four categories in a random order (*Figure 1A*). Images of all categories appeared at equal probability and no images were repeated (*Stigliani et al., 2015*). Infants participated in five conditions, which varied by the target category. We used the EEG-SSVEP approach because: (i) it affords a high signal-to-noise ratio with short acquisitions making it effective for infants (*de Heering and Rossion, 2015*; *Farzin et al., 2012*), (ii) it has been successfully used to study responses to faces in infants (*de Heering and Rossion, 2015*; *Farzin et al., 2012*; *Rekow et al., 2021*), and (iii) it enables measuring both general visual response to images by examining responses at the image presentation frequency (4.286 Hz) and category-selective responses by examining responses at the category frequency (0.857 Hz, *Figure 1A*).

As the EEG-SSVEP paradigm is novel and we are restricted in the amount of data we can obtain in infants, we first tested if we can use this paradigm and a similar amount of data to detect

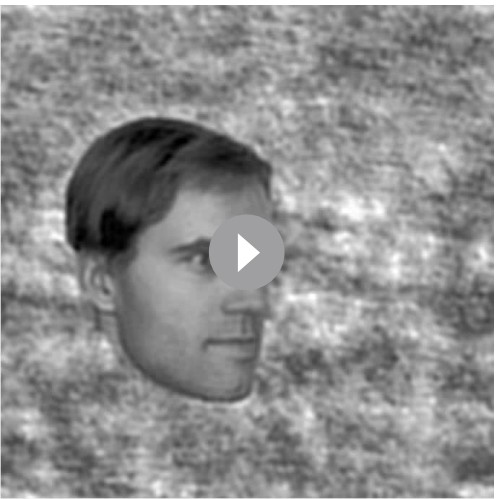

**Video 1.** Movie showing Gaussian low-pass filtered face stimuli shown in the experiment at 5 cycles per degree (cpd).

https://elifesciences.org/articles/100260/figures#video1

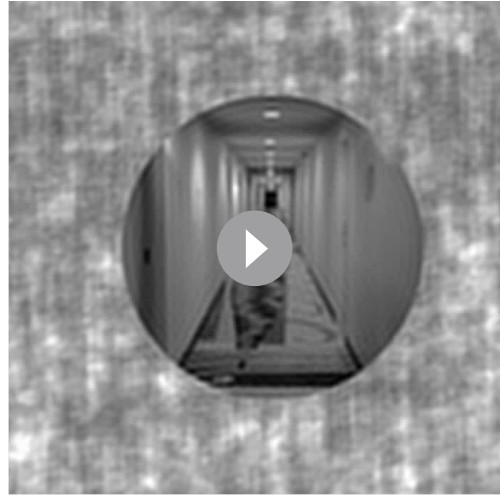

**Video 3.** Movie showing Gaussian low-pass filtered corridor stimuli shown in the experiment at 5 cycles per degree (cpd).

https://elifesciences.org/articles/100260/figures#video3

category-selective responses in adults. Results in adults validate the SSVEP paradigm for measuring category selectivity: as they show that (i) category-selective responses can be reliably measured using EEG-SSVEP with the same amount of data as in infants (*Appendix 1—figure 1*, *Appendix 1—figure 2*), and that (ii) category information from distributed spatiotemporal response patterns can be decoded with the same amount of data as in infants (*Appendix 1—figure 3*).

As infants have lower cortical visual acuity, we also tested if the stimuli are distinguishable to infants. Thus, we simulated how they may look to infants by filtering the images to match the cortical acuity of 3-month-olds (*Appendix 1—figure 4*). Despite being blurry, images of different categories are readily distinguishable by adults (*Videos 1–5*), suggesting that there is sufficient visual information in the lower spatial frequencies of the stimuli for infants to distinguish visual categories.

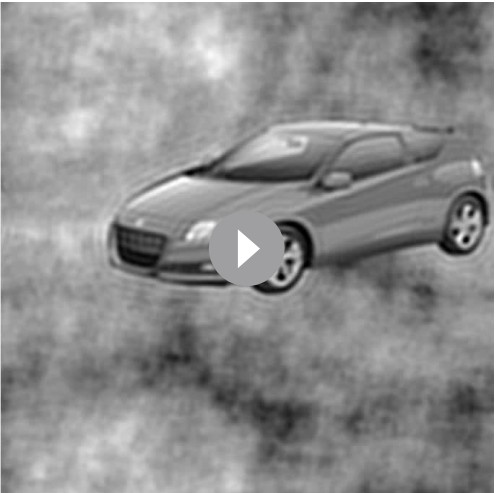

**Video 2.** Movie showing Gaussian low-pass filtered limb stimuli shown in the experiment at 5 cycles per degree (cpd).

https://elifesciences.org/articles/100260/figures#video2

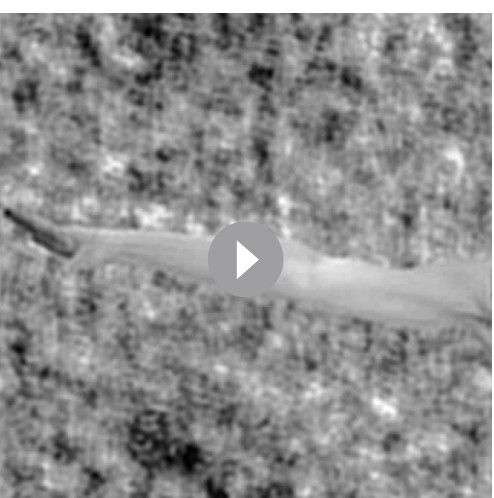

**Video 4.** Movie showing Gaussian low-pass filtered character stimuli shown in the experiment at 5 cycles per degree (cpd).

https://elifesciences.org/articles/100260/figures#video4

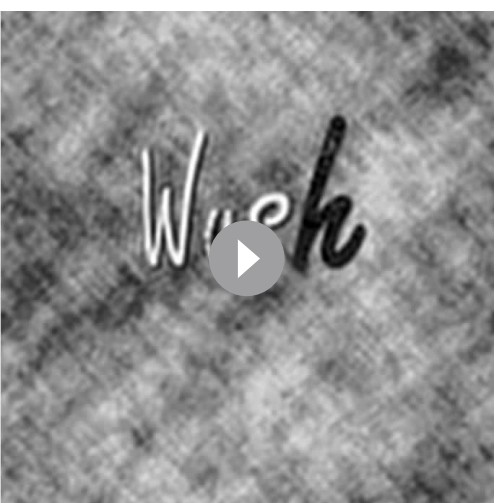

**Video 5.** Movie showing Gaussian low-pass filtered car stimuli shown in the experiment at 5 cycles per degree (cpd).

https://elifesciences.org/articles/100260/figures#video5

## Robust visual responses in occipital regions to visual stimuli in all infant age groups

We first tested if there are significant visual responses to our stimuli in infants' brains by evaluating the amplitude of responses at the image presentation frequency (4.286 Hz) and its first three harmonics. We found that in all age groups, visual responses were concentrated spatially over occipital electrodes (*Figure 2A–D*, left panel, *Appendix 1—figure 1*). Quantification of the mean visual response amplitude over a region of interest (ROI) spanning nine electrodes over early visual cortex (occipital ROI) revealed significant responses in all infant age groups at the image frequency and its first three harmonics (response amplitudes significantly above zero with false discovery rate [FDR] corrected at four levels; except for the first harmonic at 8.571 Hz in 6- to 8-month-olds; *Figure 2A–D*, right panel). Analysis of visual responses separately by category condition revealed that visual responses were not significantly different across category conditions (*Appendix 1—figure 5*; no significant main effect of category, $\beta_{category}$ = 0.08, 95% CI: –0.08–0.24, $t_{(301)}$ = 0.97, p=0.33, or category by age interaction, $\beta_{category \times age}$ = -0.04, 95% CI: –0.11–0.03, $t_{(301)}$ = –1.09, p=0.28, linear mixed model (LMM) on response amplitude to 4.286 Hz and its first three harmonics). We also tested if experimental noise varied across age groups. Noise level was estimated in the occipital electrodes by measuring the amplitude of response in frequencies up to 8.571 Hz excluding image presentation frequencies (4.286 Hz and harmonics) and category frequencies (0.857 Hz and harmonics) as this frequency range includes the relevant main harmonics. We found no significant difference in noise across age groups (*Figure 2E*). These analyses indicate that infants were looking at the stimuli as there are significant visual responses even in the youngest 3- to 4-month-old infants' and there are no significant differences in noise levels across infants of different ages.

Prior EEG data (*Conte et al., 2020*; *Taylor et al., 1999*) suggest that the timing and waveform of visual responses may vary across development. To complement the frequency domain analysis, we transformed the responses at image frequency and its harmonics to the time domain using an inverse Fourier transformation for two reasons. First, the time domain provides access to information about response timing and waveform that is not directly accessible from an analysis of responses of individual harmonics. Second, the total visual response is better reflected in the time domain as the individual harmonic amplitudes can sum constructively.

We observed that during the 233 ms image presentation, temporal waveforms had two deflections in 3- to 4-month-olds (one negativity and one positivity, *Figure 2F*) and four deflections for infants older than 4 months (two minima and two maxima, *Figure 2F*). To evaluate developmental effects, we examined the latency and amplitude of the peak visual response during two time windows related to the first deflection (60–90 ms), and the second deflection (90–160 ms for 3- to 4-month-olds, and 90–110 ms for other age groups). In general, we find that the latency of the peak deflection decreased from 3 to 15 months (*Figure 2G and H*). As data includes both cross-sectional and longitudinal measurements and we observed larger development earlier than later in infancy, we used an LMM to model peak latency as a function of the logarithm of age (see Methods). Results reveal that the latency of the peak deflection significantly and differentially decreased with age in the two time windows ($\beta_{age \times time\ window}$ = –45.78, 95% CI: –58.39 to –33.17, $t_{(118)}$ = –7.19, p=6.39 × 10$^{-11}$; LMM with age and time window as fixed effects, and participant as a random effect, all stats in *Appendix 1—table 4*, *Appendix 1—table 5*). There were larger decreases in the peak latency in the second than first time window (*Figure 2G and H*, first: $\beta_{age}$ = –7.44, 95% CI: –13.82 to –1.06, $t_{(118)}$ = –2.33, p$_{FDR}$<0.05; second: $\beta_{age}$ = –46.91, 95% CI: –56.56 to –37.27, $t_{(59)}$ = –9.73, p$_{FDR}$<0.001). Peak amplitude also differentially

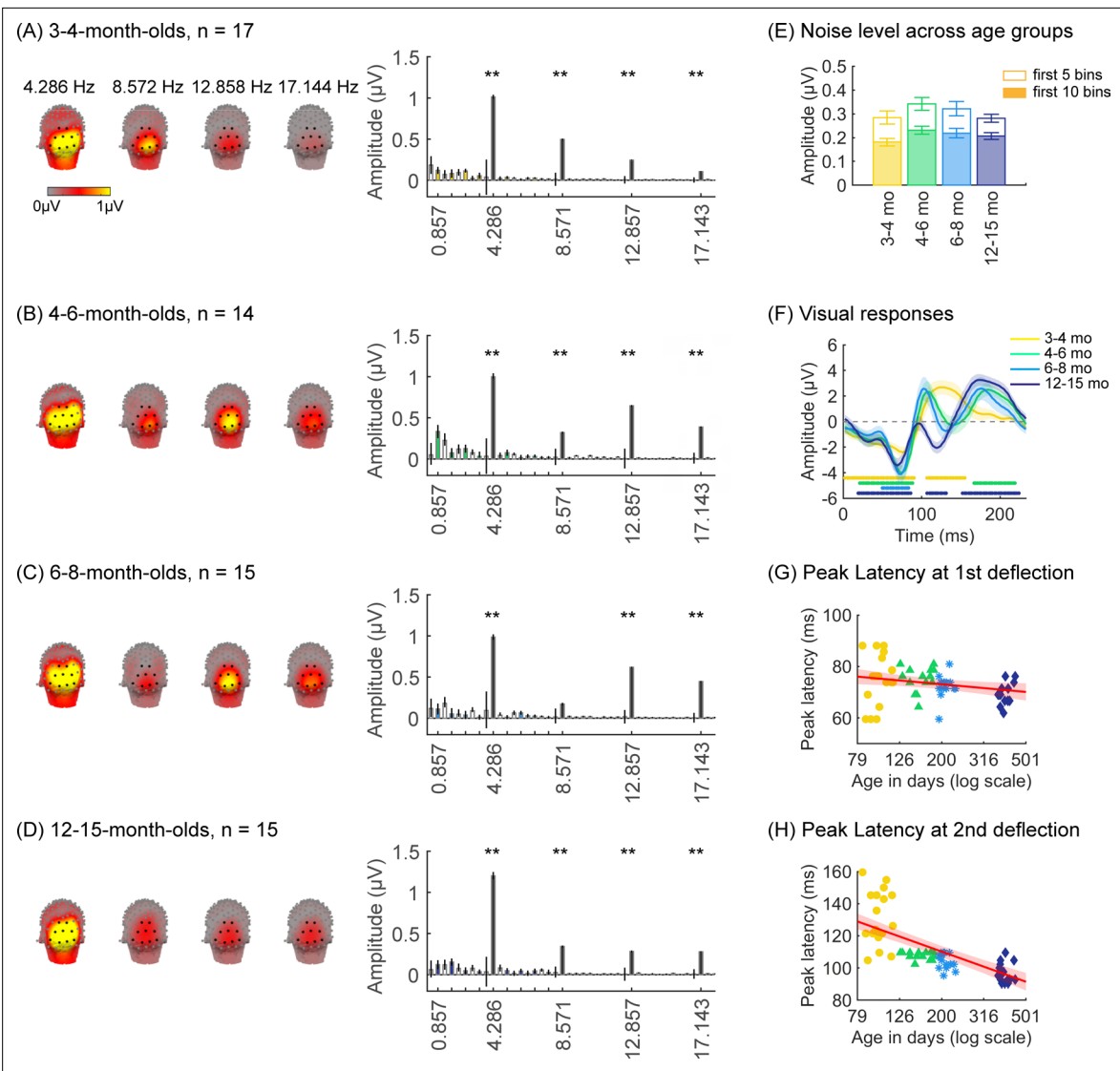

**Figure 2.** Strong visual responses over occipital cortex at the image-update frequency and its harmonics in all age groups. Each panel shows mean responses across infants in an age group. (**A**) 3- to 4-month-olds, n=17; (**B**) 4- to 6-month-olds, n=14; (**C**) 6- to 8-month-olds, n=15; (**D**) 12- to 15-month-olds, n=15. *Left panels in each row:* spatial distribution of the visual response at the image-update frequency and its first three harmonics. *Middle panels in each row:* mean Fourier amplitude spectrum across nine occipital electrodes of the occipital region of interest (ROI) showing high activity at harmonics of the image-update frequency marked out by thicker lines. Data are first averaged in each participant and each condition and then across participants. *Error bars:* standard error of the mean across participants. *Asterisks:* response amplitudes significantly larger than zero, p<0.01, false discovery rate (FDR) corrected. *Colored bars:* amplitude of response at category frequency and its harmonics. *White bars:* amplitude of response at noise frequencies. (**E**) Noise amplitudes in the frequency range up to 8.571 Hz (except for the visual response frequencies and visual category response frequencies) from the amplitude spectra in (**A**) for each age group (white bars on the spectra). *Error bars:* standard error of the mean across participants. (**F**) Mean image-update response over occipital electrodes for each age group. Waveforms are cycle averages over the period of the individual image presentation time (233 ms). *Lines:* mean response. *Shaded areas:* standard error of the mean across participants of each group. *Horizontal lines colored by age group:* significant responses vs. zero (p<0.05 with a cluster-based analysis, see Methods). (**G**) Peak latency for the first peak in the 60–90 ms interval after stimulus onset. Each dot is a participant; dots are colored by age group. *Line:* linear mixed model (LMM) estimate of peak latency as a function of log10(age). *Shaded area:* 95% confidence interval (CI). (**H**) Same as (**G**) but for the second peak in the 90–160 ms interval for 3- to 4-month-olds, and 90–110 ms for older infants.

develops across the two windows ($\beta_{age \times time\ window}$ = –4.90, 95% CI: –8.66 to –1.14, $t_{(118)}$ = –2.58, p=0.01, *Appendix 1—table 6*, *Appendix 1—table 7*). The decrease in peak amplitude with age was significant only for the second deflection ($\beta_{age}$ = –3.59, 95% CI: –6.38 to –0.81, $t_{(59)}$ = –2.58, p=0.01, LMM). These data suggest that the temporal dynamics of visual responses over occipital cortex develop from 3 to 15 months of age.

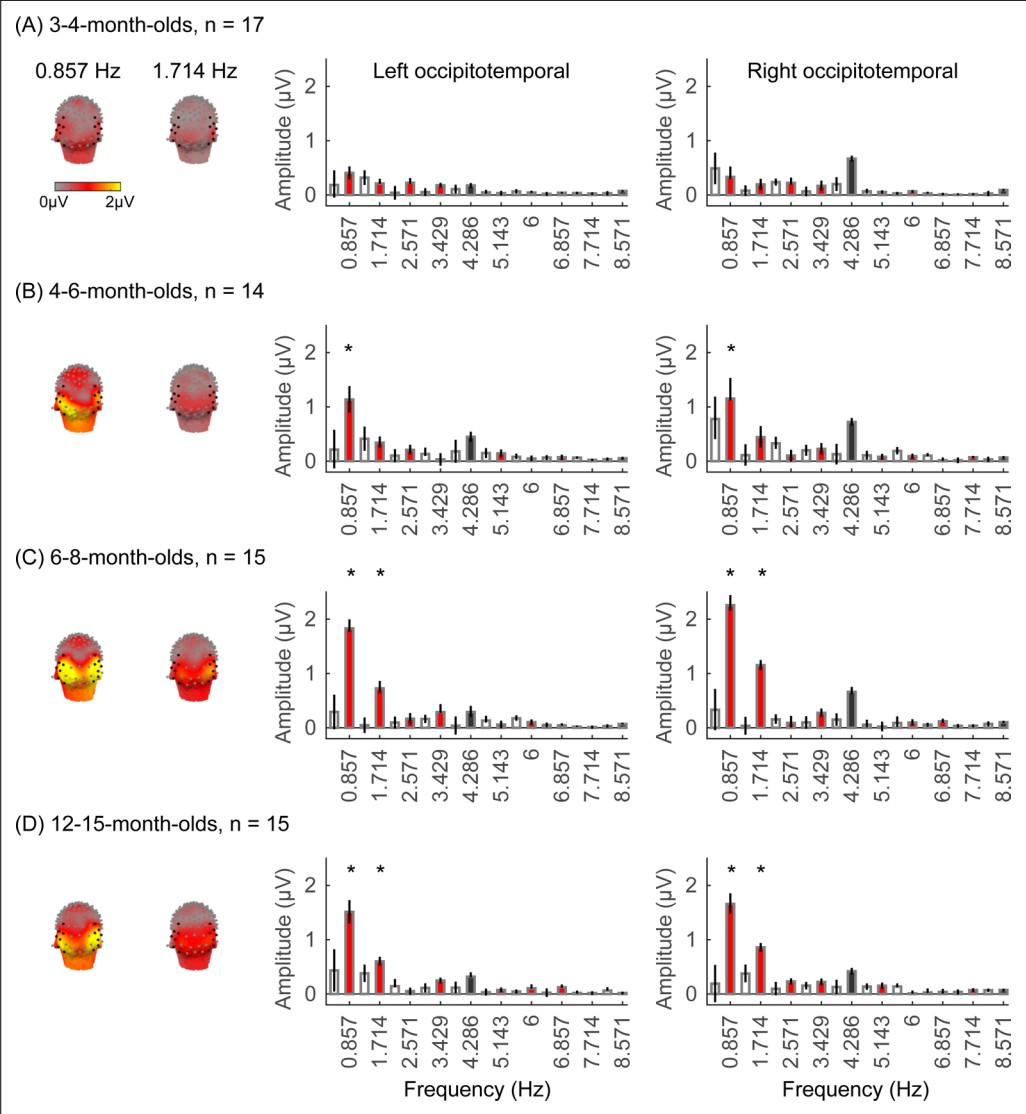

**Figure 3.** Face responses emerge over occipitotemporal electrodes after 4 months of age. Each panel shows mean responses at the category frequency and its harmonics across infants in an age group. (**A**) 3- to 4-month-olds, n=17; (**B**) 4- to 6-month-olds; n=14; (**C**) 6- to 8-month-olds, n=15; (**D**) 12- to 15-month-olds, n=15. Left panels in each row: spatial distribution of response to category frequency at the 0.857 Hz and its first harmonic. Harmonic frequencies are indicated on the top. Right two panels in each row: mean Fourier amplitude spectrum across two regions of interest (ROIs): seven left and seven right occipitotemporal electrodes (shown in black on the left panel). Data are first averaged across electrodes in each participant and then across participants. Error bars: standard error of the mean across participants of an age group. Asterisks: significant amplitude vs. zero (p<0.05, false discovery rate [FDR] corrected at two levels). Black bars: image frequency and harmonics; colored bars: category frequency and harmonics. White bars: noise frequencies. Responses for the other categories (limbs, corridors, characters, and cars) in *Appendix 1—figures 5–8*.

## What is the nature of category-selective responses in infants?

We next examined if in addition to visual responses to the rapid image stream, there are also category-selective responses in infants, by evaluating the amplitude of responses at the category frequency (0.857 Hz) and its harmonics. This is a selective response as it reflects the relative response to images of category above the general visual response. *Figure 3* shows the spatial distribution and amplitude of the mean category response for faces and its harmonics in each age group. Mean category-selective responses to limbs, cars, corridors, and words are shown in *Appendix 1—figures 6–9*. We

analyzed mean responses over two ROIs spanning seven electrodes each over the left (LOT) and right occipitotemporal (ROT) cortex where high-level visual regions are located (*Xie et al., 2019*).

We found significant group-level category responses to some but not all categories and a differential development of category-selective responses during infancy. The largest and earliest developing category-selective responses were to faces. In contrast to visual responses, which were centered over occipital electrodes (*Figure 2A–D*, left panel), significant categorical responses to faces (at 0.857 Hz and its first harmonic, 1.714 Hz) were observed over lateral occipitotemporal electrodes (*Figure 3A–D*, left panel). Notably, there were significant responses to faces over bilateral occipitotemporal electrodes in 4- to 6-month-olds at 0.857 Hz (*Figure 3B*, response amplitudes significantly above zero with Hotelling's T2 statistic, $p_{FDR}<0.05$, FDR corrected over two levels: the category frequency and its first harmonic), as well as 6- to 8-month-olds and 12- to 15-month-olds at the category frequency and its first harmonic (*Figure 3C and D*, both $p_{FDR}<0.05$). However, there were no significant responses to faces in 3- to 4-month-olds at either the category frequency or its harmonics (*Figure 3A*, right panel). These data suggest that face-selective responses start to reliably emerge over lateral occipitotemporal cortex between 4 and 6 months of age.

We did not find significant group-level category-selective responses that survived FDR correction to any of the other categories before 6 months of age (*Appendix 1—figures 6–9*, except for a weak but statistically significant response for cars in the ROT ROI in 3- to 4-month-olds). Instead, we found significant category-selective responses that survived FDR correction for (i) limbs in 6- to 8-month-olds in the ROT ROI (*Appendix 1—figure 6*), (ii) corridors in 6- to 8-month-olds and 12- to 15-months-old in the left occipitotemporal (LOT) ROI (*Appendix 1—figure 7*), and (iii) characters in 6- to 8-month-olds in the ROT ROI, and in 12- to 15-month-olds in bilateral occipitotemporal ROI (*Appendix 1—figure 8*).

We next examined the development of the category-selective responses separately for the right and left lateral occipitotemporal ROIs. The response amplitude was quantified by the root mean square (RMS) amplitude value of the responses at the category frequency (0.857 Hz) and its first harmonic (1.714 Hz) for each category condition and infant. With an LMM analysis, we found significant development of response amplitudes in both the occipitotemporal ROIs which varied by category (LOT ROIs: $\beta_{category \times age} = -0.21$, 95% CI: –0.39 to –0.04, $t_{(301)} = -2.40$, $p_{FDR}<0.05$; ROT ROIs: $\beta_{category \times age} = -0.26$, 95% CI: –0.48 to –0.03, $t_{(301)} = -2.26$, $p_{FDR}<0.05$, LMM as a function of log (age) and category; participant: random effect).

We evaluated the temporal dynamics of category-selective waveforms by transforming the data at the category frequency and its harmonics to the time domain. This analysis was done separately for each of the LOT and ROT ROIs for each category and age group. Consistent with frequency domain analyses, average temporal waveforms over lateral-occipital ROIs show significant responses to faces that emerge at ~4 months of age (*Figure 4A*, significant responses relative to zero, cluster-based nonparametric permutation 10,000 times, two-tailed t-test, at p<0.05). The temporal waveforms of responses to faces in infants show an initial positive deflection peaking ~500 ms after stimulus onset followed by a negative deflection peaking at ~900 ms. Notably, mean waveforms associated with limbs, corridors, and characters in lateral occipital ROIs are different from faces: there is only a single negative deflection that peaks at ~500 ms after stimulus onset, which is significant only in 6- to 8- and 12- to 15-month-olds (*Figure 4B–D*). There was no significant category response to cars in infants except for a late (~1000 ms) positive response in 4- to 6-month-olds (*Figure 4E*). These results show that both the timing and waveform differ across categories, which suggests that there might be additional category information in the distributed spatiotemporal response.

We next examined the development of the peak response and latency of the category waveforms separately for the right and left lateral occipitotemporal ROIs. We found significant development of the peak response in the right lateral occipitotemporal ROI which varied by category ($\beta_{category \times age} = -1.09$, 95% CI: –2.00 to –0.14, $t_{(301)} = -2.26$, $p_{FDR}<0.05$, LMM as a function of log(age) and category; participant: random effect). Post hoc analyses revealed that the peak response for faces significantly increased from 3 to 15 months (*Figure 4A* right, $\beta_{age} = 7.27$, 95% CI: 4.03–10.51, $t_{(59)} = 4.50$, $p_{FDR}<0.05$, LMM as a function of log(age); participant: random effect) and the peak response for limbs significantly decreased (*Figure 4B* right, $\beta_{age} = -2.90$, 95% CI: –5.41 to –0.38, $t_{(59)} = -2.31$, p=0.02, not significant after FDR correction over five category levels). There were no other significant developments of peak amplitude (*Appendix 1—table 8*, *Appendix 1—table 9*).

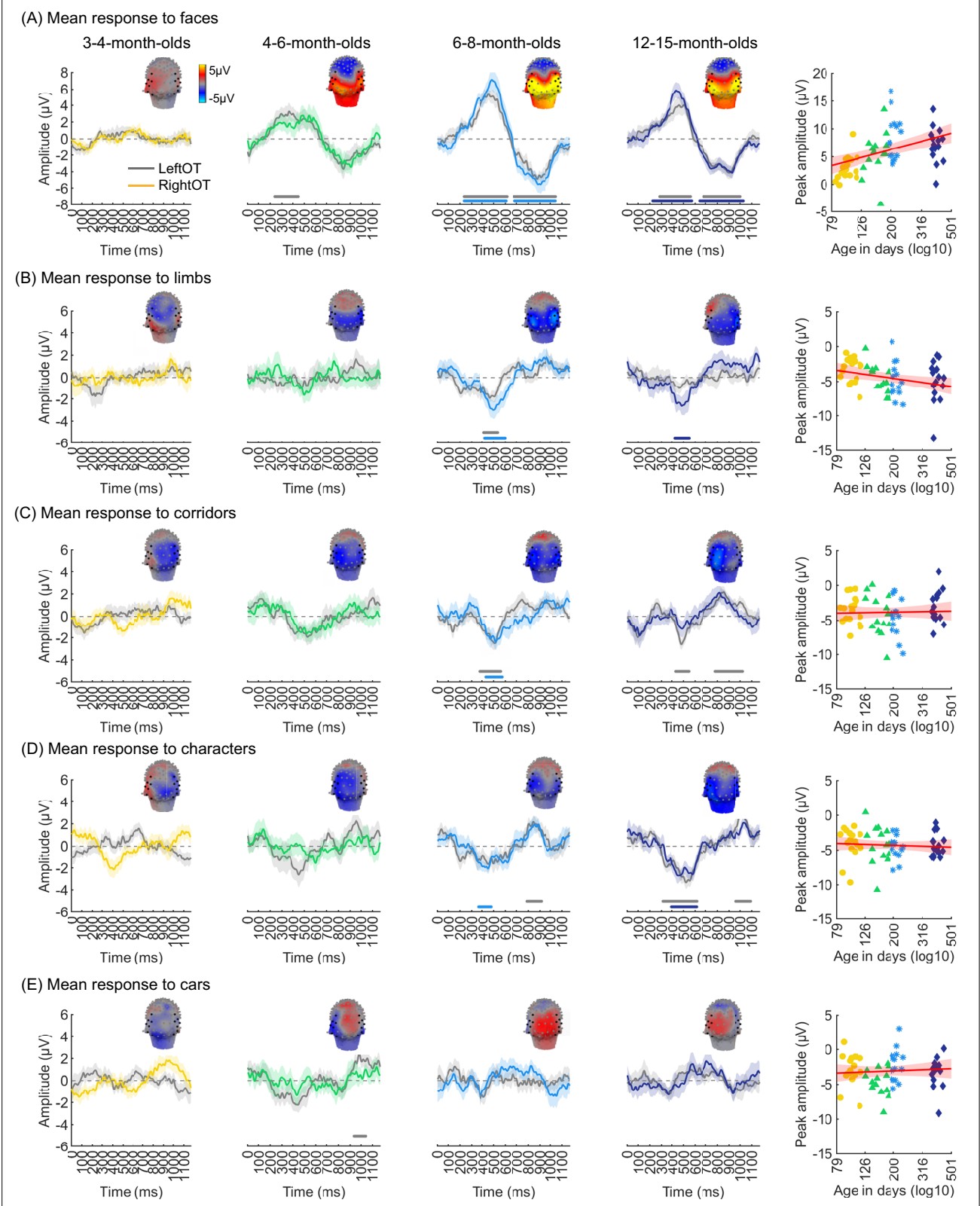

**Figure 4.** Temporal dynamics of category-selective responses as a function of age. Category-selective responses to (**A**) faces, (**B**) limbs, (**C**) corridors, (**D**) characters, and (**E**) cars over left and right occipitotemporal region of interest (ROI). Data are averaged across electrodes of an ROI and across individuals. Left four panels in each row show the responses in the time domain for the four age groups. *Colored lines:* mean responses in the right occipitotemporal ROI. *Gray lines:* mean responses in the left occipitotemporal ROI. *Colored horizontal lines above x-axis:* significant responses relative

Figure 4 continued

to zero for the right OT ROI. *Gray horizontal lines above x-axis:* significant responses relative to zero for the left OT ROI. *Top:* 3D topographies of the spatial distribution of the response to target category stimuli at a 483–500 ms time window after stimulus onset. *Right panel in each row:* amplitude of the peak deflection defined in a 400–700 ms time interval after stimulus onset. Each dot is a participant; dots are colored by age group. *Red line:* linear mixed model (LMM) estimate of peak amplitude as a function of log10(age). *Shaded area:* 95% CI.

Additionally, for all categories, the latency of the peak response in the ROT ROI significantly decreased from 3 to 15 months of age ($\beta_{age}$ = –173.17, 95% CI: –284.73 to –61.61, $t_{(301)}$ = –3.05, p=0.002, LMM as a function of log(age) and category; participant: random effect). We found no significant development of peak latency in the LOT ROI (*Appendix 1—table 8*, *Appendix 1—table 9*).

## Are spatiotemporal patterns of responses to visual categories consistent across infants?

As we observed different mean waveforms over the lateral occipital ROIs for the five categories (*Figure 4*), we asked whether the distributed spatiotemporal patterns of brain responses evoked by each category are unique and reliable across infants. We reasoned that if different categories generated consistent distributed spatiotemporal responses, an independent classifier would be able to predict the category an infant was viewing from their distributed spatiotemporal pattern of response. Thus, we used a leave-one-out-cross-validation (LOOCV) approach (*Figure 5A*) and tested if

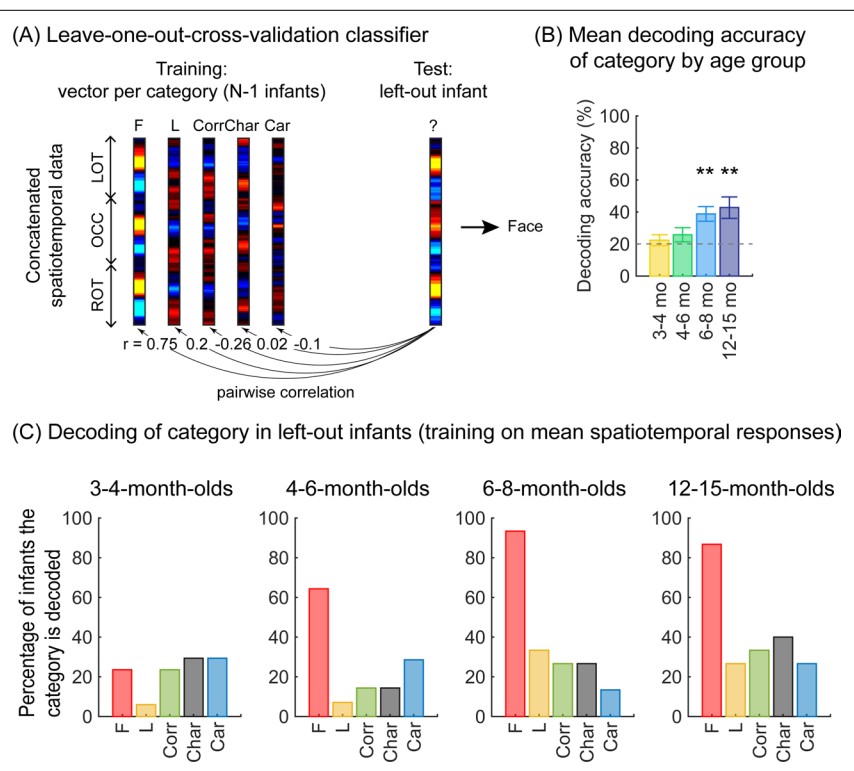

**Figure 5.** Successful decoding of faces from mean spatiotemporal responses starting from 4 months of age. (**A**) An illustration of winner-takes-all leave-one-out-cross-validation (LOOCV) classifier from mean spatiotemporal response patterns of each category. Spatiotemporal patterns of response for each category are generated by concatenating the mean time courses from N–1 infants from three regions of interest (ROIs): left occipitotemporal (LOT), occipital (OCC), and right occipitotemporal (ROT). At each iteration, we train the classifier with the mean spatiotemporal patterns of each category from N–1 infants, and test how well it predicts the category the left-out infant is viewing from their spatiotemporal brain response. The winner-take-all (WTA) classifier determines the category based on the training vector that has highest pairwise correlation with the test vectors. (**B**) Mean decoding accuracies across all five categories in each age group. *Asterisks:* significant decoding above chance level (p<0.01, Bonferroni corrected, one-tailed). (**C**) Percentage of infants in each age group we could successfully decode for each category. *Dashed lines:* chance level.

a classifier can decode the category a left-out infant viewed based on the similarity of their distributed spatiotemporal response to the mean response to each of the categories in the remaining N–1 infants. We calculated for each infant the mean category waveform (same as *Figure 4*) across the occipital and lateral occipitotemporal ROIs and concatenated the waveforms across the three ROIs to generate the distributed spatiotemporal response to a category (*Figure 5A*). The classifier was trained and tested separately for each age group.

Results reveal two main findings. First, the LOOCV classifier decoded category information from brain responses significantly above the 20% chance level in infants aged 6 months and older but not in younger infants (*Figure 5B*, 6- to 8-month-olds, significant above chance: $t_{(14)} = 4.1$, $p_{FDR}<0.01$, one-tailed, FDR corrected over four age groups; 12- to 15-month-olds, $t_{(14)} = 3.4$, $p_{FDR}<0.01$). This suggests that spatiotemporal patterns of responses to different categories become reliable across infants after 6 months of age. Second, examination of classification by category shows that the LOOCV classifier successfully determined from spatiotemporal responses when infants were viewing faces in 64% of 4- to 6-month-olds, in 93% of 6- to 8-month-olds, and 87% of 12- to 15-month-olds (*Figure 5C*). In contrast, classification performance was numerically lower for the other categories (successful classification in less than 40% of the infants). This suggests that a reliable spatiotemporal response to faces that is consistent across infants develops after 4 months of age and dominates classification performance.

## What is the nature of categorical spatiotemporal patterns in individual infants?

While the prior analyses leverage the power of averaging across electrodes and infants, this averaging does not provide insight to fine-grained neural representations within individual infants. To examine the finer-grain representation of category information within each infant's brain, we examined the distributed spatiotemporal responses to each category across the 23 electrodes spanning the LOT and ROT cortex in each infant. We tested: (i) if categorical representations in an infant's brain are reliable across different images of a category, and (ii) if category representations become more distinct during the first year of life. We predicted that if representations become more similar across items of a category and more dissimilar between items of different categories then category distinctiveness (defined as the difference between mean within and between category similarity) would increase from 3 to 15 months of age.

To examine the representational structure, we calculated representation similarity matrices (RSMs) across odd/even split-halves of the data in each infant. Each cell in the RSM quantifies the similarity between two spatiotemporal patterns: On-diagonal cells of the RSM quantify the similarity of distributed spatiotemporal responses to different images from the same category and off-diagonal cells quantify the similarity of spatiotemporal responses to images from different categories. Categorical structure will manifest in RSMs as positive on diagonal values indicating reliable within-category spatiotemporal responses which are higher than off-diagonal between category similarity (*Figure 6*, *Appendix 1—figure 3B*, and *Appendix 1—figure 10*).

Examination of mean RSMs in each age group reveals no reliable category information in individuals at 3- to 4-month-olds or 4- to 6-month-olds, as within-category similarity is not significantly above zero (*Figure 6A* and 3- to 4-month-olds: on-diagonal, –0.03 ±0.06, p=0.96, one-tailed; 4- to 6-month-olds: on-diagonal: 0.009 ± 0.11, p=0.38). However, starting around 6 months some category structure emerges in the RSMs. In particular, distributed responses to faces become reliable as within category similarity for faces is significantly above zero in 6- to 8-month-olds (*Figure 6A*, 0.31 ± 0.24, $t_{(14)} = 5.1$, $p_{FDR}<0.05$, FDR corrected over five category levels), and stays reliable in 12- to 15-month-olds (*Figure 6A*, 0.26 ± 0.24, $t_{(14)} = 4.18$, $p_{FDR}<0.05$). Distributed responses to limbs become reliable later on as within category similarity for limbs is significantly above zero in 12- to 15-months-olds (*Figure 6A*, 0.11 ± 0.21, $t_{(14)} = 1.98$, p=0.03, but not surviving FDR correction at five levels).

Next, we evaluated the development of category distinctiveness, which was calculated for each category and infant. Individual infants' category distinctiveness is shown in *Figure 6B* (infants ordered by age) and in the scatterplots in *Figure 6C*. In infants younger than 4 months (120 days) category distinctiveness is largely close to zero or even negative, suggesting no differences between spatiotemporal responses to one category vs. another. Category distinctiveness increases with age and

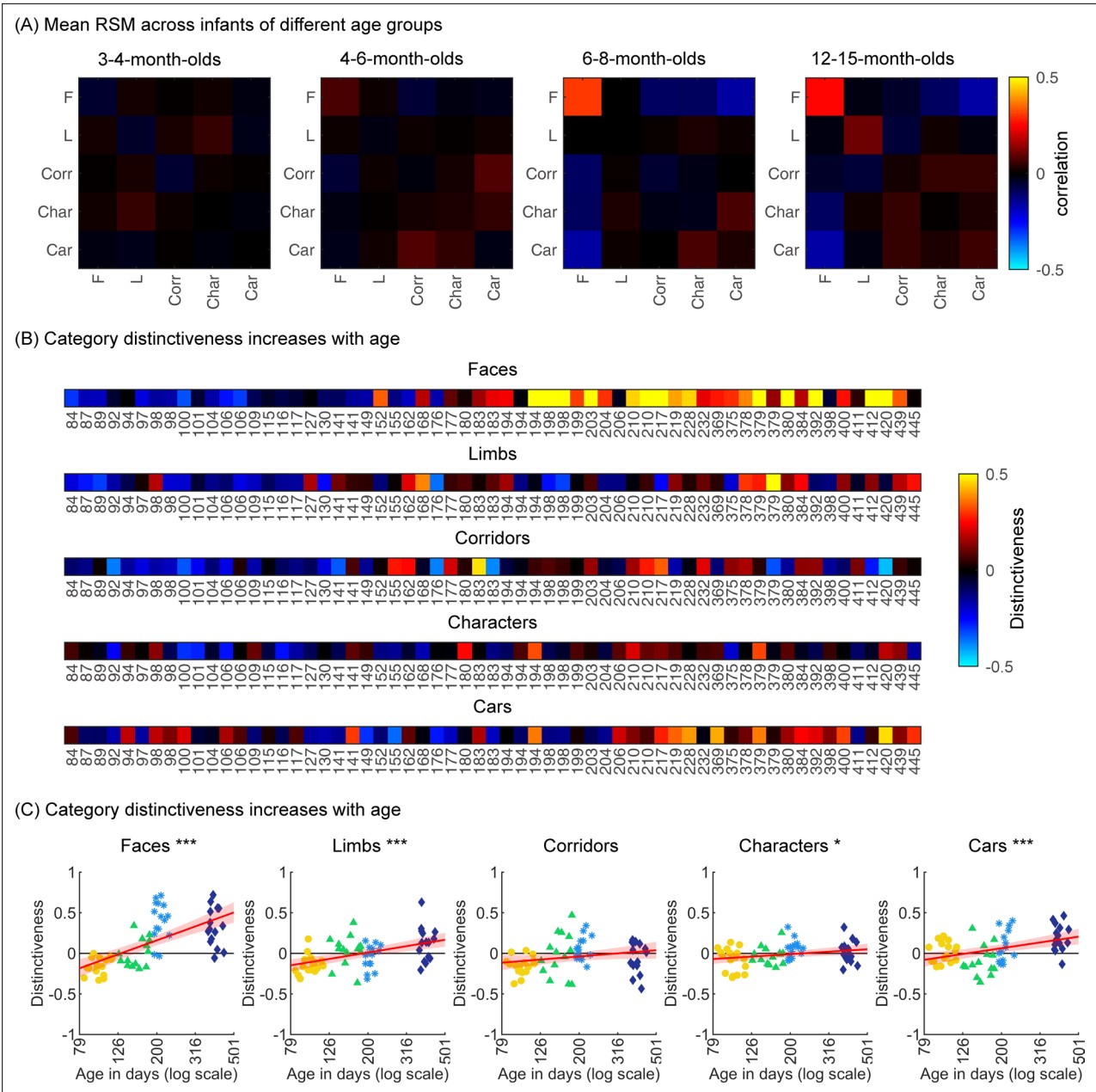

**Figure 6.** Individual split-half spatiotemporal pattern analyses reveal category information slowly emerges in the visual cortex after 6 months of age. (**A**) Representation similarity matrices (RSMs) generated from odd/even split-halves of the spatiotemporal patterns of responses in individual infants. Spatiotemporal patterns for each category are generated by concatenating the mean time courses of each of 23 electrodes across left occipitotemporal (LOT), occipital (OCC), and right occipitotemporal (ROT). (**B**) Category distinctiveness calculated for each infant and category by subtracting the mean between-category correlation values from the within-category correlation value. (**C**) Distinctiveness as a function of age; panels by category; each dot is a participant. Dots are colored by age group. *Red line:* linear mixed model (LMM) estimates of distinctiveness as a function of log10(age). *Shaded area:* 95% CI.

becomes more positive from 84 to 445 days of age (*Figure 6B and C*). The biggest increase is for faces where after ~6 months of age (194 days) face distinctiveness is consistently positive in individual infants (13/15 infants aged 194–364 days and 12/15 infants aged 365–445 days). The increase in distinctiveness is more modest for other categories and appears later in development. For example, positive distinctiveness for limbs and cars in individual infants is consistently observed after 12 months of age (*Figure 6B and C*; limbs: 9/15 infants aged 365–445 days vs. 5/15 infants aged 194–364 days; cars: 12/15 365–445 days vs. 7/15 194–364 days).

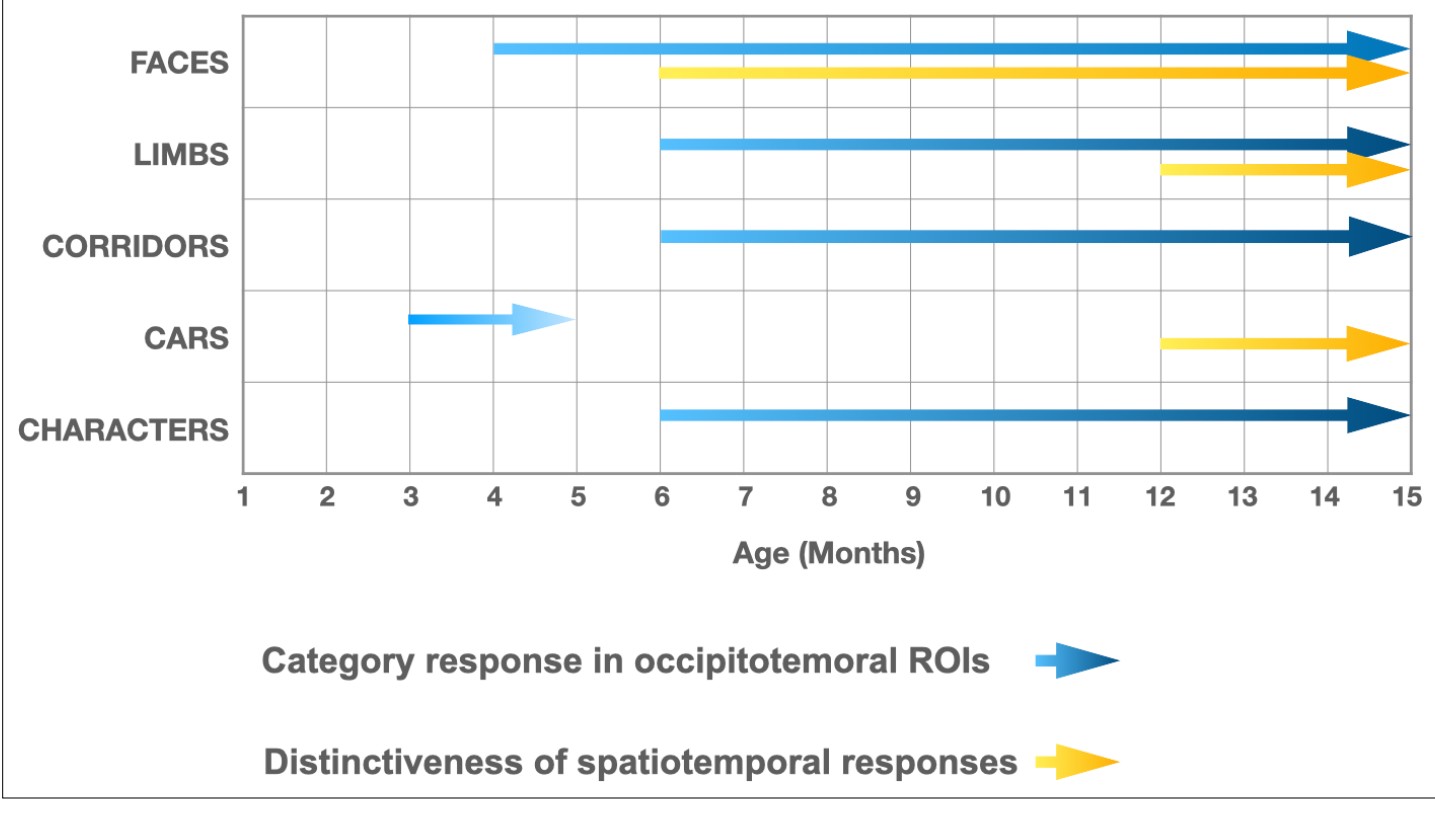

**Figure 7.** Selective responses to items of a category and distinctiveness in distributed patterns develop at different times during the first year of life. *Blue arrows*: presence of significant mean region of interest (ROI) category-selective responses in lateral occipital ROIs, combining results of analyses in the frequency and time domains. *Yellow arrows*: presence of significantly above zero distinctiveness in the distributed spatiotemporal response patterns across occipital and lateral occipital electrodes.

Using LMMs we determined if distinctiveness significantly changed with age (log transformed) and category (participant, random factor). Results indicate that category distinctiveness significantly increased from 3 to 15 months of age ($\beta_{\beta\beta age}$ = 0.77, 95% CI: 0.54–1.00, $t_{(301)}$ = 6.62, p=1.67×10$^{-10}$), and further that development significantly varies across categories ($\beta_{age \times category}$ = –0.13, 95% CI: –0.2 to –0.06, $t_{(301)}$ = –3.61, p=3.5×10$^{-4}$; main effect of category, $\beta_{category}$ = 0.27, 95% CI: 0.11–0.43, $t_{(301)}$ = 3.38, p=8.2×10$^{-4}$). Post hoc analyses for each category (*Figure 6C*) reveal that distinctiveness significantly increased with age for faces ($\beta_{age}$ = 0.9, 95% CI: 0.6–1.1, $t_{(59)}$ = 6.8, $p_{FDR}$<0.001), limbs ($\beta_{age}$ = 0.4, 95% CI: 0.2–0.6, $t_{(59)}$ = 5.0, $p_{FDR}$<0.001), characters ($\beta_{age}$ = 0.2, 95% CI: 0.02–0.3, $t_{(59)}$ = 2.2, $p_{FDR}$<0.05), and cars ($\beta_{age}$ = 0.4, 95% CI: 0.2–0.5, $t_{(59)}$ = 3.7, $p_{FDR}$<0.001). Post hoc t-tests show that for faces, the category distinctiveness is significantly above zero after 6 months of age (6- to 8-month-olds, $t_{(14)}$ = 6.73, $p_{FDR}$<0.05; 12- to 15-month-olds, $t_{(14)}$ = 5.30, $p_{FDR}$<0.05) and for limbs and cars at 12–15 months of age (limbs: $t_{(14)}$ = 2.19, $p_{FDR}$<0.05; cars: $t_{(14)}$ = 4.53, $p_{FDR}$<0.05). This suggests that category distinctiveness slowly emerges in the visual cortex of infants from 3 to 15 months of age, with the largest and fastest development for faces.

## Discussion

We find that both selective responses to items of a category over others across lateral occipital ROIs and the distinctiveness of distributed visual category representations progressively develop from 3 to 15 months of age. Notably, we find a differential development of category-selective responses (*Figure 7*), whereby responses to faces emerge the earliest, at 4–6 months of age and continue to develop through the first year of life. Category-selective responses to limbs, corridors, and characters follow, emerging at 6–8 months of age. Our analysis of the distinctiveness of the distributed spatiotemporal patterns to each category also finds that distributed representations to faces become more robust in 6- to 8-month-olds and remain robust in 12- to 15-month-olds. While the distinctiveness of

distributed patterns to limbs and cars only become reliable at 12–15 months of age. Together these data suggest a rethinking of the development of category representations during infancy as they not only suggest that category representations are learned, but also that representations of categories that may have innate substrates such as faces, bodies, and places emerge at different times during infancy.

## Reliable category representations start to emerge at 4 months of age

While 3- to 4-month-old infants have significant and reliable evoked visual responses over early visual cortex, we find no reliable category representations of faces, limbs, corridors, or characters in these young infants. Both analyses of average responses across lateral occipital ROIs and analyses of distributed spatiotemporal responses across visual cortex find no reliable category representations in 3- to 4-month-olds, either when examining mean response across an ROI or in distributed spatiotemporal patterns across visual cortex. The earliest categorical responses we find are for faces, and they emerge at 4–6 months of age.

Is it possible that there are some category representations in 3- to 4-month-olds, but we lack the sensitivity to measure them? We believe this is unlikely, because (i) we can measure significant visual responses from the same 3- to 4-month-olds, (ii) with the same amount of data, we can measure category-selective responses and decode category information from distributed spatiotemporal responses in infants older than 4 months and in adults. As using SSVEP to study high-level representations is a nascent field (*Gentile and Rossion, 2014*; *Retter et al., 2020*; *Peykarjou, 2022*), future work can further examine how SSVEP parameters such as stimulus and target category presentation rate may affect the sensitivity of measurements in infants (see review by *Peykarjou, 2022*).

Our findings together with a recent fMRI study in 2- to 10-month-olds (*Kosakowski et al., 2022*) provide accumulating evidence for multiple visual categories representations in infants' brains before the age of one. However, there are also differences across studies. The earliest we could find reliable group-level category-selective responses for faces was 4- to 6-month-olds and for limbs and corridors only after 6 months of age. In contrast, *Kosakowski et al., 2022*, report category-selective responses to faces, bodies, and scenes in example 4- to 5-month-olds. Group average data in their study found significant face- and place-selective responses in infants' VTC but not in LOTC, and significant body-selective responses in LOTC, but not VTC. Because *Kosakowski et al., 2022*, report group-averaged data across infants spanning 8 months, their study does not provide insights to the time course of this development. We note that, the studies differ in several ways: (i) measurement modalities (fMRI in *Kosakowski et al., 2022*, and EEG here), (ii) the types of stimuli infants viewed: in *Kosakowski et al., 2022*, infants viewed isolated, colored, and moving stimuli, but in our study, infants viewed still, gray-level images on phase-scrambled backgrounds, which were controlled for several low-level properties, and (iii) contrasts used to detect category-selective responses, whereby in *Kosakowski et al., 2022*, the researchers identified within predefined parcels – the top 5% of voxels that responded to the category of interest vs. objects, here we contrasted the category of interest vs. all other categories the infant viewed. Thus, future research is necessary to determine whether differences between findings are due to differences in measurement modalities, stimulus format, and data analysis choices.

## Face representations emerge around 4–6 months of age

Recognizing faces (e.g. a caregiver's face) is crucial for infants' daily lives. Converging evidence from many studies suggest that infants have significant and reliable face-selective neural responses at 4–6 months of age (*Farzin et al., 2012*; *de Heering and Rossion, 2015*; *Deen et al., 2017*; *Guy et al., 2016*; *Halit et al., 2004*). While some studies report responses to face-like (high-contrast paddle-like) stimuli in newborns (*Johnson et al., 1991*; *Buiatti et al., 2019*; *Goren et al., 1975*) and significant visual evoked responses to faces in 3-month-olds (*Halit et al., 2003*; *Cassia et al., 2006*; *Peykarjou, 2022*; *Tzourio-Mazoyer et al., 2002*), these studies have largely compared responses to an isolated face vs. another isolated object. In contrast, we do not find reliable face-selective responses (*Figures 3–4*) or reliable distributed representations (*Figures 5–6*) to faces in 3- to 4-month-olds when responses to faces are contrasted to many other items and when stimuli are shown on a background rather than in isolation. Our findings are consistent with longitudinal research in macaques showing that robust cortical selectivity to faces takes several months to emerge (*Livingstone et al., 2017*) and

support the hypothesis that experience with faces is necessary for the development of cortical face selectivity (*Arcaro et al., 2019*; *Livingstone et al., 2017*; *Arcaro et al., 2017*).

Our data also reveal that face-selective responses and distributed representations to faces become more robust in 6- to 8-month-olds and remain robust in 12- to 15-month-olds. For example, successful decoding of faces in the group level was observed in 80% of individual infants based on several minutes of EEG data. Reliable distributed spatiotemporal responses to different images of faces become significantly different from responses to images from different categories. This robust decoding has important clinical ramifications as it may serve as an early biomarker for cortical face processing, which is important for early detection of social and cognitive developmental disorders such as Autism (*Rossion, 2020*; *Vettori et al., 2019*) and Williams syndrome (*Farran et al., 2020*). Future research is necessary for elucidating the relationship between the development of brain responses to faces to infant behavior. For example, it is interesting that at 6 months of age, when we find robust face representations, infants also start to exhibit recognition of familiar faces (like parents) and stranger anxiety (*Kobayashi et al., 2020*).

One fascinating aspect of the development of cortical face selectivity is that among the categories we tested, selectivity to faces seems to emerge the earliest at around 4 months of age, yet the development of selectivity and distributed representations to faces is protracted compared to objects and places (*Golarai et al., 2007*; *Peelen et al., 2009*; *Scherf et al., 2007*). Indeed, in both our data and prior work, face-selective responses and distributed representations to faces in infants are immature compared to adults (*Conte et al., 2020*; *Xie et al., 2022*), and a large body of work has shown that face selectivity (*Nordt et al., 2021*; *Golarai et al., 2007*; *Scherf et al., 2007*; *Taylor et al., 1999*; *Peelen et al., 2009*; *Cantlon et al., 2011*; *Cohen et al., 2019*) and distributed representations to faces (*Nordt et al., 2023*) continue to develop during childhood and adolescence. This suggests that not only experience during infancy but also life-long experience with faces, sculpts cortical face selectivity. We speculate that the extended cortical plasticity for faces may be due to both the expansion of social circles (family, friends, and acquaintances) across the lifespans and also the changing statistics of faces we socialize with (e.g. child and adult faces have different appearance).

## A new insight about cortical development: different category representations emerge at different times during infancy

To our knowledge, our study is the first to examine the development of both ROI level and spatiotemporal distributed responses in infants across the first year of life. We note that both analyses find that category information to faces develops before other categories. However, there are also some differences across analyses (*Figure 7*). For example, for limbs and corridors we find significant category-selective responses at the ROI level in lateral occipitotemporal ROIs starting at 6–8 months but no reliable distinct distributed responses across visual cortex at this age. In contrast, for cars, we find an opposite pattern where there is a distinct spatiotemporal pattern in 12- to 15-month-olds even as there is no significant car-selective response in the ROI level. As these approaches have different sensitivities, they reveal insights to the nature of the underlying representations. For example, as visible in *Figure 4*, limbs and corridor have a clear category-selective waveform in both in 6- to 8- and 12- to 15-months-olds, but the waveform of limbs and its spatial distribution is not that different from that to corridors, which may explain why distinctiveness of spatiotemporal patterns for limbs is low in 6- to 8-month-olds (*Figure 6*). Likewise, even as there is no significant response for cars (*Figure 4e*), its spatiotemporal pattern is consistently different than for other categories giving rise to a distinctive spatiotemporal response by 12 months (*Figure 6*).

In sum, the key finding from our study is that the development of category selectivity during infancy is non-uniform: face-selective responses and representations of distributed patterns develop before representations to limbs and other categories. We hypothesize that this differential development of visual category representations may be due to differential visual experience with these categories during infancy. This hypothesis is consistent with behavioral research using head-mounted cameras that revealed that the visual input during early infancy is dense with faces, while hands become more prevalent in the visual input later in development and especially when in contact with objects (*Fausey et al., 2016*; *Jayaraman et al., 2017*). Additionally, a large body of research has suggested that young infants preferentially look at faces and face-like stimuli (*Spriet et al., 2022*; *Mondloch et al., 1999*; *Pascalis et al., 1995*; *Johnson et al., 1991*), as well as look longer at faces than other objects (*Fausey*

*et al., 2016*), indicating that not only the prevalence of faces in babies' environments but also longer looking times may drive the early development of face representations. Further supporting the role of visual experience in the formation of category selectivity is a study that found that infant macaques that are reared without seeing faces do not develop face selectivity but develop selectivity to other categories in their environment like body parts (*Arcaro et al., 2017*). An alternative hypothesis is that differential development of category representations is of maturational origin. For example, we found differences in the temporal dynamics of visual responses among four infant age groups, which suggests that the infant's visual system is still developing during the first year of life. While underlying maturational mechanisms are yet unknown they may include myelination and cortical tissue maturation (*Grotheer et al., 2022*; *Natu et al., 2021*; *Tooley et al., 2021*; *Adibpour et al., 2024*; *Lebenberg et al., 2019*; *Gilmore et al., 2018*). Future studies can test experience-driven vs. maturational alternatives by examining infants' visual diet, looking behavior, and anatomical brain development and examine responses using additional behaviorally relevant categories such as food (*Bannert and Bartels, 2022*; *Jain et al., 2023*; *Pennock et al., 2023*). These measurements can test how environmental and individual differences in visual experiences may impact infants' developmental trajectories. Specifically, a visual experience account predicts that differences in visual experience would translate into differences in brain development, but a maturational account predicts that visual experience will have no impact on the development of category representations.

Together our findings not only suggest that visual experience is necessary for the development of visual category representations, including faces, but also necessitate a rethinking of how visual category representations develop in infancy. Moreover, this differential development during infancy is evident even for categories that have evolutionary importance and may have innate substrates such as faces, bodies, and places (*Kanwisher, 2010*; *Sugita, 2009*; *Mahon and Caramazza, 2011*; *Bi et al., 2016*). Finally, our findings have important ramifications for theoretical and computational models of visual development as well as for the assessment of atypical infant development.

## Methods

### Participants

Ethical permission for the study was obtained from the Institutional Review Board of Stanford University. Parents of the infant participants provided written informed consent prior to their first visit and also prior to each session if they came for multiple sessions. Participants were paid 20$/hr for participation. Participants were recruited via ads on social media (Facebook and Instagram).

Sixty-two full-term, typically developing infants were recruited. Twelve participants were part of an ongoing longitudinal study that obtained both anatomical MRI and EEG data in infants. Some of the infants participated in both studies and some only in one of the studies. Infants were recruited at around newborn, 3 months, 6 months, and 12 months. We did not recruit infants between 8 and 12 months of age because around 9 months there is little contrast between gray and white matter in anatomical MRI scans that were necessary for the MRI study. Infants came for several sessions spanning ~3 months apart (seven 3- to 4-month-olds, three 4- to 6-month-olds, eight 6- to 8-month-olds, and twelve 12- to 15-month-olds). Data from 19 infants (nine 3- to 4-month-olds, six 4- to 6-month-olds, and eight 6- to 8-month-olds; among whom seven were longitudinal) were acquired in two visits within a 2-week span to obtain a sufficient number of valid data epochs. *Appendix 1—table 1* contains participants' demographic information (sex and race). The youngest infants were 3 months of age, as the EEG setup requires the infants to be able to hold their head and look at the screen in front of them. 23 adults (14 females) also participated in the study. All participants had normal/corrected-to-normal vision and provided written informed consent.

Data *exclusion criteria:* We excluded participants who had less than 20 valid epochs (1.1667 s/epoch) per category, had noise/muscular artifacts during the EEG recordings, couldn't record data, or had no visual responses over the occipital electrodes. As such, we excluded (i) five infants due to an insufficient number of epochs, (ii) two infants who had no visual responses, (iii) ten infants due to technical issues during data collection, and (iv) three adults due to excessive noise/muscular artifacts during EEG. In total, we report data from 45 infants (*Appendix 1—table 1*) and 20 adults (13 females, 19–38 years) that met inclusion criteria.

## Visual stimuli

Natural grayscale images of adult faces, limbs, corridors, characters, and cars are used as stimuli, with 144 images per category from the fLOC image database (https://github.com/VPNL/fLoc, copy archived at *Yan, 2024*; *Stigliani et al., 2015*). The size, view, and retinal position of the items varied, and the items were overlaid on phase-scrambled backgrounds that were generated from a randomly drawn image in the stimulus set. The images were also controlled for multiple low-level differences between stimuli of different categories including their luminance, contrast, similarity, and spatial frequency power distributions using the SHINE toolbox (*Willenbockel et al., 2010*). As only five of ten categories from *Stigliani et al., 2015*, were used, we evaluated the stimuli used in our experiments to test if they differed in (i) contrast, (ii) luminance, (iii) similarity, and (iv) spatial frequency. Results show that categories were largely matched on most metrics (*Figure 1B* and Appendix). The stimuli were presented on a gamma-corrected OLED monitor screen (SONY PVM-2451; SONY Corporation, Tokyo Japan) at a screen resolution of 1920 ×1080 pixels and a monitor refresh rate of 60 Hz. When viewed from 70 cm away, each image extended a visual angle of approximately 12°.

## EEG protocol

The experiments were conducted in a calm, dimly illuminated lab room. Stimuli were presented using custom stimulus presentation software with millisecond timing precision. During testing, infant participants were seated on their parent's laps in front of the screen at a distance of 70 cm. One experimenter stood behind the presentation screen to monitor where the infant was looking. The visual presentation was paused if the infant looked away from the screen and was continued when the infant looked again at the center of the screen. To motivate infants to fixate and look at the screen, we presented at the center of the screen small (~1°) colored cartoon images such as butterflies, flowers, and ladybugs. They were presented in random order with durations uniformly distributed between 1 and 1.5 s. For adults, we used a fixation cross of the same size instead of the cartoons and asked the participants to fixate and indicate when the fixation's color changed by pressing a space bar key on a keyboard. EEG measurements for infant participants continued until the infant no longer attended the screen and we obtained between 2 and 12 different 70 s sequences per individual. For adult participants, we acquired 12 sequences per individual.

A frequency-tagging paradigm (*de Heering and Rossion, 2015*; *Farzin et al., 2012*) was used to measure brain responses. In the experiment, randomly selected images from five categories were presented sequentially at a rate of 4.286 Hz (~233 ms per image) with no inter stimulus interval during each 70 s sequence. For each condition, one category was determined as the target category; for this category random selected images from that category were presented first and followed by four images randomly drawn from the other four categories with no regular order (*Figure 1A*). The target images are therefore presented periodically at 0.857 Hz (i.e. 4.286 Hz/5), but the intervals between sequential presentations of images from the other four categories was not periodic. The probability of image occurrences across categories was equal at 20%. The experiment had five conditions, one for each of the following target categories: faces, limbs, corridors, characters, and cars. Each 70 s experimental sequence was composed of five 14 s long conditions which included a 1.1667 s stimulus fade-in and a 1.1667 s stimulus fade-out. The order of the category conditions was randomized within each 70 s sequence. No image was repeated within a sequence. Two presentation frequencies were embedded in the experiment: (i) the image frequency (4.286 Hz), which is predicted to elicit visual responses to all stimuli over occipital visual cortex, and, (ii) the category frequency (0.857 Hz), which is predicted to elicit a category-selective response over lateral occipital-temporal electrodes.

## EEG acquisition

EEG data were recorded at 500 Hz from a 128-channel EGI High-Density Geodesic Sensor Net. For infants, the net was connected to a NetAmps 300 amplifier (Electrical Geodesics, Inc, Eugene, OR, USA). For the adults, the net was connected to a NetAmps400 amplifier. The EEG recording was referenced online to a single vertex (electrode Cz) and the channel impedance was kept below 50 KΩ.

## Pre-processing

EEG recordings were down-sampled to 420 Hz and were filtered using a 0.03–50 Hz bandpass filter with custom signal processing software. Artifact rejection was performed in two steps. For infants, first,

channels with more than 20% of samples exceeding a 100–150 µV amplitude threshold were replaced with the average amplitude of its six nearest-neighbor channels. The continuous EEG signals were then re-referenced to the common average of all channels and segmented into 1166.7 ms epochs (i.e. duration of five stimuli starting with a target category image followed with four images drawn from the rest four categories). Epochs with more than 15% of time samples exceeding threshold (150–200 µV) were excluded further on a channel-by-channel basis (*Norcia et al., 2017*). For adults, the two-step artifact rejection was performed with different criteria as EEG response amplitudes are lower in adults than infants (*Norcia et al., 2017*). EEG channels with more than 15% of samples exceeding a 30 µV amplitude threshold were replaced by the average value of their neighboring channels. Then the EEG signals were re-referenced to the common average of all channels and segmented into 1.1667 s epochs. Epochs with more than 10% of time samples exceeding threshold (30–80 µV) were excluded on a channel-by-channel basis (*Kohler et al., 2020*).

*Appendix 1—table 2* shows the number of epochs (1.1667 s each) we acquired before and after data pre-processing summing across all five categories. We used data after pre-processing for further analyses. There was no significant difference in the number of pre-processed epochs across infant age groups ($F_{(3,57)}$ = 1.5, p=0.2). The number of electrodes being interpolated for each age group were 10.0±4.8 for 3- to 4-month-olds, 9.9 ± 3.7 for 4- to 6-month-olds, 9.9 ±3.9 for 6- to 8-month-olds, and 7.7 ±4.7 for 12- to 15-month-olds. There was no significant difference in the number of electrodes being interpolated across infant age groups ($F_{(3,55)}$ = 0.78, p=0.51).

## Univariate EEG analyses

Both image-update and categorical EEG visual responses are reported in the frequency and time domain over three ROIs: two occipito-temporal ROIs (LOT: channels 57, 58, 59, 63, 64, 65, and 68; ROT channels: 90, 91, 94, 95, 96, 99, and 100) and one occipital ROI (channels 69, 70, 71, 74, 75, 76, 82, 83, and 89). These ROIs were selected a priori based on a previously published study (*Xie et al., 2019*). We further removed several channels in these ROIs for two reasons: (i) Three outer rim channels (i.e. 73, 81, and 88) were not included in the occipital ROI for further data analysis for both infant and adult participants because they were consistently noisy. (ii) Three channels (66, 72, and 84) in the occipital ROI, one channel (50) in the LOT ROI, and one channel (101) in the ROT ROI were removed because they did not show substantial responses in the group-level analyses.

### Frequency domain analysis

Individual participant's pre-processed EEG signals for each stimulus condition were averaged over two consecutive epochs (2.3334 s). The averaged time courses for each participant were then converted to the frequency domain at a frequency resolution of 0.4286 Hz via a discrete Fourier transform (DFT). The frequency bins of interest are at exactly every other bin in the frequency spectrum. The real and imaginary Fourier coefficients for each of the categorical and image-update responses for each condition were averaged across participants (vector averaging) to obtain a group-level estimate. The amplitudes of response were then computed from the coherently averaged vector. Hotelling's T2 statistic (*Victor and Mast, 1991*) was used to test whether response amplitudes were significantly different from zero. We used Benjamini's & Hochberg's FDR procedure to correct for multiple comparisons.

### Image-update visual responses (image frequency)

The amplitude and phase of the evoked response at the image presentation rate and its first three harmonics (4.286 Hz, 8.571 Hz, 12.857 Hz, and 17.143 Hz).

### Categorical responses (category frequency)

The amplitude and phase of the response at the category repetition frequency and its second harmonic (0.857 Hz, 1.714 Hz) for each category condition.

### Time domain analyses

Pre-processed time domain EEG signals of each participant were low-passed filtered with a 30 Hz cut-off. The raw EEG signals contain many components including categorical responses (0.857 Hz and harmonics), general visual responses (4.286 Hz and harmonics), and noise. To separate out the temporal waveforms of these two responses, we first transformed the epoch-averaged (2.3334 s)

time series of each condition into frequency domain using a DFT. Then, we used an inverse DFT to transform back to the time domain keeping only the responses at the category frequency and its harmonics, zeroing the other frequencies. The same approach was used to separate the visual responses by using an inverse DFT transform of the responses at 4.286 Hz and harmonics.

## Categorical responses
We kept responses at frequencies of interest (0.857 Hz and its harmonics up to 30 Hz, excluding the harmonic frequencies that overlapped with the image frequency and its harmonics) and zeroed responses in other frequencies. Then we applied an inverse Fourier transform to transform the data to the time domain. We further segmented the time series into 1.1667 s epochs and averaged across these epochs for each condition and individual. The mean and standard error across participants were computed for each condition at each time point.

## Image-update visual responses
A similar procedure was performed except that frequencies of interest are 4.286 Hz and its harmonics, and the rest were zeroed. As temporal waveforms for image-update responses were similar across different category conditions, we averaged waveforms across all five conditions and report the mean response (*Figure 2*).

## Statistical analyses
To determine time windows in which amplitudes were significantly different from zero for each condition, we used a cluster-based nonparametric permutation t-test, 10,000 permutations, with a threshold of p<0.05, two-tailed on the post-stimulus onset time points (0–1167 ms) (*Appelbaum et al., 2006*; *Blair and Karniski, 1993*). The null hypothesis is that the evoked waveforms are not different from zero at any time point. For each permutation, we assigned random signs to the data of individual participants and computed the group-level difference against zero using a t-test. We then calculated the cluster-level statistic as the sum of t-values in the consecutive time points with p-values less than 0.05 (*Maris and Oostenveld, 2007*). We calculated the maximum cluster-level statistic for each permutation to generate a nonparametric reference distribution of cluster-level statistics. We rejected the null hypothesis if the cluster-level statistic for any consecutive time points in the original data was larger than 97.5% or smaller than 2.5% of the values in the null distribution.

## Decoding analyses
### Group level
We used an LOOCV classifier to test if spatiotemporal responses patterns to each of the five categories were reliable across participants. The classifier was trained on averaged data from N–1 participants and tested on how well it predicted the category the left-out participant was viewing from their brain activations. This procedure was repeated for each left-out participant. We calculated the averaged category temporal waveform for each category across channels of our three ROIs: seven LOT, nine occipital, and seven ROT, as the exact location of the channels varies across individuals. Then, we concatenated the waveform from the three ROIs to form a spatiotemporal response vector (*Figure 5A*). At each iteration, the LOOCV classifier computed the correlation between each of the five category vectors from the left-out participant (test data, for an unknown stimulus) and each of the mean spatiotemporal vectors across the N–1 participants (training data, labeled data). The winner-take-all classifier classifies the test vector to the category of the training vector that yields the highest correlation with the training vector (*Figure 5A*). For a given test pattern, correct classification yields a score of 1 and an incorrect classification yields a score of 0. For each left-out infant, we computed the percentage correct across all categories, and then the mean decoding performance across all participants in an age group (*Figure 5B*).

### Individual level
Similar to group level with two differences: (i) All analyses were done within an individual using a split-half approach. That is, the classifier was trained on one half of the data (i.e. odd or even trials) and tested on the other half of the data. (ii) Spatiotemporal patterns for each category used the

**Table 1.** Linear mixed models (LMMs).

| Variable | LMM formula | Results |
|---|---|---|
| Peak latency: latency of the peak waveform in each time window; window 1: 60–90 ms or window 2: 90–160 ms for 3- to 4-month-olds, and 90–110 ms for other age groups | Peak latency ~1 + log10(age in days)×time window + (1\|participant) | *Figure 2G*. *Appendix 1—table 4* |
| | Peak latency ~1 + log10(age in days) + (1\|participant) | *Figure 2H*. *Appendix 1—table 5* |
| Category-selective response amplitude: root mean square (RMS) of category-selective response at category frequency (0.857 Hz) and its first harmonic (1.714 Hz) | Response amplitude ~1 + log10(age in days)×category + (1\|participant) | |
| Peak amplitude: peak response in a 400–700 ms time window | Peak amplitude ~1 + log10(age in days)×category + (1\|participant) | *Appendix 1—table 7* |
| | Peak amplitude ~1 + log10(age in days) + (1\|participant) | *Figure 4*. *Appendix 1—table 8* |
| Category distinctiveness of spatiotemporal responses for each of the five categories | Category distinctiveness ~log10(age in days)×category + (1\|participant) | |
| | Category distinctiveness ~log10(age in days) + (1\|participant) | *Figure 6B* |

concatenated waveforms across 23 channels spanning the occipital and bilateral occipitotemporal ROIs.

## Category distinctiveness (*Nordt et al., 2023*)

Category distinctiveness is defined as the difference between the similarity (correlation coefficient) of spatiotemporal responses within a category across odd and even splits using different images and the average between-category similarity of spatiotemporal responses across odd and even splits (*Nordt et al., 2023*). Distinctiveness is higher when the within-category similarity is positive and the between-category similarity is negative and varies from –2 (no category information) to 2 (maximal category information). We computed category distinctiveness for each of the five categories as in each infant and determined if it varied from 3 to 15 months of age.

## Statistical analyses of developmental effects

To quantify developmental effects, we used LMMs (*Bosker and Snijders, 2011*), with the 'fitlme' function in MATLAB version 2021b (MathWorks, Inc). LMMs allow explicit modeling of both within-subject effects (e.g. longitudinal measurements) and between-subject effects (e.g. cross-sectional data) with unequal number of points per participants, as well as examine main and interactive effects of both continuous (age) and categorical (e.g. stimulus category) variables. We used random-intercept models that allow the intercept to vary across participants (term: 1|participant). In all LMMs, we measured development as a function of log 10(age in days) as development is typically faster earlier on. Indirect evidence comes from neuroimaging and post-mortem studies showing that the structural development of infants' brains is nonlinear, with development in the first 2 years being rapid, especially the first year (*Gilmore et al., 2018*).

We report slope (rate of development), interaction effects, and their significance. *Table 1* summarizes LMMs used in this study.

## Analysis of noise

To test whether EEG noise levels vary with age, e.g. whether noise in the EEG data is larger in the younger infants than older ones, we quantified the response amplitudes in the occipital ROI in the frequency domain, at frequency bins next to the category and image frequency bins (white bars in *Figure 2A–D*, right panel). The noise level was quantified as the amplitude of response up to 8.571 Hz excluding image presentation frequencies (4.286 Hz and harmonics) and category frequencies (0.857 Hz and harmonics) as this frequency range includes the relevant main harmonics (*Figure 2E*). We used a LMM to test if noise varies with age, with participant as a random effect:

$$\text{Noise amplitude} \sim 1 + \log10(\text{age in days}) + (1|\text{participant})$$

We found no significant differences in noise amplitude across infant age groups (*Figure 2E*, mean amplitude across the first five noise bins: $\beta_{age}$ = –0.005, 95% CI: –0.12 to –0.11, $t_{(59)}$ = –0.09, p=0.93; mean noise across the first 10 bins: $\beta_{age}$ = .04, 95% CI: –0.03 to –0.12, $t_{(59)}$ = 1.16, p=0.25, LMMs with age (log transformed) as fixed effects and participant as a random effect).

## Acknowledgements

This work was supported by grants from the Wu Tsai Neurosciences Institute of Stanford University, the Human Centered Artificial Intelligence Institute of Stanford University to KGS and AMN.

## Additional information

### Funding

| Funder | Grant reference number | Author |
|---|---|---|
| Wu Tsai Neurosciences Institute, Stanford University | 139471 | Anthony M Norcia Kalanit Grill-Spector |

The funders had no role in study design, data collection and interpretation, or the decision to submit the work for publication.

### Author contributions

Xiaoqian Yan, Conceptualization, Data curation, Formal analysis, Validation, Investigation, Visualization, Methodology, Writing – original draft, Writing – review and editing; Sarah Shi Tung, Bella Fascendini, Yulan Diana Chen, Data curation; Anthony M Norcia, Kalanit Grill-Spector, Conceptualization, Resources, Software, Supervision, Funding acquisition, Methodology, Writing – original draft, Writing – review and editing

### Author ORCIDs

Xiaoqian Yan (ID) https://orcid.org/0000-0003-4711-7428
Bella Fascendini (ID) https://orcid.org/0000-0002-7690-6752

### Ethics

Ethical permission (eprotocol number: 48634) for the study was obtained from the Institutional Review Board of Stanford University. Parents of the infant participants provided written informed consent prior to their first visit and also prior to each session if they came for multiple sessions.

Reviewer #1 (Public review): https://doi.org/10.7554/eLife.100260.3.sa1
Reviewer #2 (Public review): https://doi.org/10.7554/eLife.100260.3.sa2
Reviewer #3 (Public review): https://doi.org/10.7554/eLife.100260.3.sa3
Author response https://doi.org/10.7554/eLife.100260.3.sa4

## Additional files

### Supplementary files
• MDAR checklist

### Data availability

Individual preprocessed EEG data, and code for all analyses are available on Github: https://github.com/VPNL/InfantObjectCategorization (copy archived at *Yan, 2023*). Individual EEG data is also available on OSF (https://doi.org/10.17605/OSF.IO/G5FTA).

The following dataset was generated:

| Author(s) | Year | Dataset title | Dataset URL | Database and Identifier |
|---|---|---|---|---|
| Yan X | 2023 | Infant object categorization | https://doi.org/10.17605/OSF.IO/G5FTA | Open Science Framework, 10.17605/OSF.IO/G5FTA |

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

## Appendix 1

### Demographic information

**Appendix 1—table 1.** Demographic information.

|  | 3–4 months (N=17) | 4–6 months (N=14) | 6–8 months (N=15) | 12–15 months (N=15) |
|---|---|---|---|---|
| Age at test (days) | 84–117 | 127–183 | 194–232 | 369–445 |
| **Sex** |  |  |  |  |
| Female | 7 | 7 | 6 | 4 |
| Male | 10 | 7 | 9 | 11 |
| **Race** |  |  |  |  |
| White | 5 | 4 | 6 | 5 |
| Black |  |  |  |  |
| Asian |  | 1 | 1 | 2 |
| Mixed races | 11 | 8 | 6 | 7 |
| Unknown | 1 | 1 | 2 | 1 |

**Appendix 1—table 2.** Average number of valid epochs summed across all five categories for each age group before and after data pre-processing.

|  | 3–4 **months** (N=17) | 4–6 **months** (N=14) | 6–8 **months** (N=15) | 12–15 **months** (N=15) | **Adults** (N=20) |
|---|---|---|---|---|---|
| Before pre-processing | 281 (± 103) | 270 (± 86) | 346 (± 111) | 324 (± 78) | 600 (± 0) |
| After pre-processing | 223 (± 89) | 219 (± 73) | 266 (± 91) | 269 (± 77) | 560 (± 37) |
| Ratio (after/before) | 79% | 81% | 77% | 83% | 93% |

### Low-level image properties analyses

As images of items of visual categories vary both on low-level and high-level properties, and both the EEG signal and babies' attention may be affected by low-level properties like contrast or luminance, it is important to control for several low-level factors such as luminance, contrast, similarity spatial frequency, as well as high-level properties such as familiarity across categories. Here, we used a subset of categories from *Stigliani et al., 2015*, that did not contain familiar stimuli to our participants and were controlled for several low-level factors using the SHINE toolbox (*Willenbockel et al., 2010*). To validate the effect of employing SHINE, we evaluated several low-level metrics and tested if they differed across categories: (i) *Luminance:* mean gray level of each image; (ii) *contrast:* mean standard deviation of gray-level values of each image; (iii) *similarity:* mean pixel-wise similarity between gray levels across pairs of images of a category, and (iv) *spatial frequency distribution*: we transformed each image to the Fourier domain using FFT and calculated its circular amplitude spectrum. We calculated these metrics for each image, then tested if contrast, luminance, similarity metrics were significantly different across categories using pairwise nonparametric permutation t-tests (10,000 times, with Bonferroni correction). While it is difficult to completely control multiple low-level metrics across categories (*Willenbockel et al., 2010*), our analyses show that images are largely controlled for several low-level metrics across categories, and that medians, means, and ranges are matched across categories (*Figure 1B*; *Appendix 1—table 3*). With respect to differences in contrast and luminance, there are no significant differences across categories except that limbs have slightly but significantly lower luminance (ps<0.05, Bonferroni corrected) and lower pairwise similarity (ps<0.01) and higher contrast (ps<0.05, except for corridors) than other categories. For similarity, the medians and ranges are similar across categories, but the outliers vary producing significant differences across categories (permutation tests, p<0.05, Bonferroni corrected). For spatial frequency distribution, the means and standard deviations are similar across categories (*Figure 1B*, right). We further ran Kolmogrov-Smirnov tests for each pair of categories with Bonferroni correction, our results revealed significantly different distributions across category pairs (ps<0.001, except for limbs and corridors).

**Appendix 1—table 3.** Mean (± SD) values of contrast, luminance, and similarity metrics across images within each five stimuli categories.

| | Faces | Limbs | Corridors | Characters | Cars |
|---|---|---|---|---|---|
| Contrast | 0.469 (2.533e-4) | 0.391 (3.91e-4) | 0.404 (4.04e-4) | 0.2856 (2.856e-4) | 0.3269 (3.269e-4) |
| Luminance | 0.6733 (0.0007) | 0.6729 (0.0011) | 0.6732 (0.0011) | 0.6732 (0.0008) | 0.6733 (0.0009) |
| Similarity | 0.9855 (0.0063) | 0.9555 (0.0039) | 0.9569 (0.0051) | 0.9563 (0.0033) | 0.9570 (0.0044) |

## Analyses of adult data

In the study, one concern is that the amount of empirical data typically collected in infants is less than in adults, which may compromise the experimental power to detect responses in infants as it may affect the noise bandwidth (sensitivity) of the frequency tagging analysis. Thus, we tested whether it is possible to measure category-selective responses in 20 adults using the same experiment (*Figure 1*) and the same amount of data collected in infants using three types of analyses. By using the same number of trials in adults as those obtained from our infant participants for data analyses, our goal was to test that we had sufficient power to detect categorical responses from infants using the experimental paradigm. We expect temporal and amplitude differences between adults and infants (*Xie et al., 2022*) as infants have immature brains and skulls. For example, cortical gyrification, which determines the orientation of the electrical fields generated in a certain part of the brain region on the scalp, still undergoes development during the first 2 years of infants' lives (*Li et al., 2014*). Second, adults' skulls are thicker and have lower conductivity than infants' skulls, thus electrical signals on their scalp are lower than infants (*Grieve et al., 2003*). Nonetheless, we tested if we could in principle detect category information in adults with the same amount of data as infants. We reasoned that if category information can be detected in adults and signals are stronger in infants then we should have the power to detect category information in infants.

### Visual responses

We averaged the pre-processed data across two consecutive epochs (2.3334 s) and across conditions for each individual to measure visual responses to all images that were presented at 4.286 Hz and then transformed the data to the frequency domain. *Appendix 1—figure 1A* shows the three-D topographies of group-averaged responses at 4.286 Hz and its first three harmonics (left panel) and the group-averaged Fourier amplitude spectrum over middle occipital and occipitotemporal ROIs (right panel). We used Hotelling's T2 statistic to test the amplitude significance and found significant visual responses in adults at 4.286 Hz over the occipital ROI containing nine electrodes over early visual cortex (p<0.05, FDR corrected at four levels) and significant visual responses at 4.286 Hz and its first two harmonics over the occipitotemporal ROI (14 electrodes) (ps<0.05).

Examining the temporal waveform of the visual response (filtered at the image frequency and its harmonics), we found a relatively slow waveform with one peak, after 100 ms since stimulus onset (*Appendix 1—figure 1B*).

(A) adult visual responses in the frequency domain, n = 20

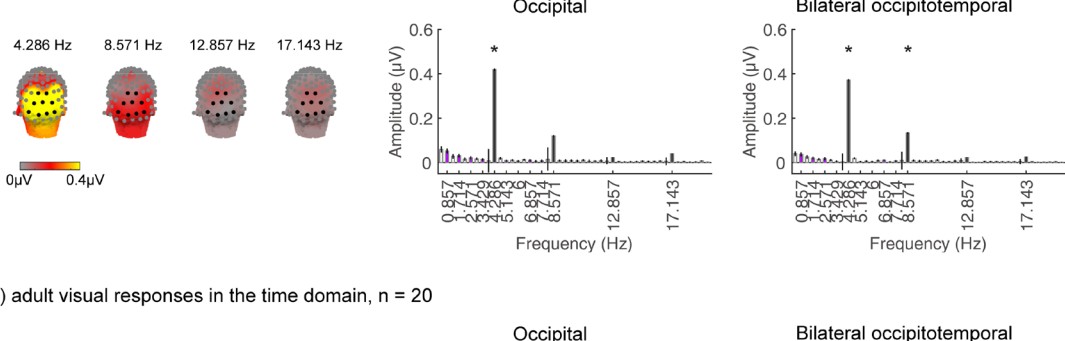

**Appendix 1—figure 1.** Robust visual and categorical responses recorded over occipitotemporal and occipital cortex in 20 adults. (**A**) *Left panel:* spatial distribution of visual response at 4.286 Hz and harmonic. Harmonic frequencies are indicated on the top. *Right panel:* mean Fourier amplitude spectrum across 14 electrodes in the occipitotemporal and 9 electrodes in the occipital regions of interest (ROIs). *Error bars:* standard error of the mean across participants. *Black bars:* image frequency and harmonics; Purple bars: category frequency and harmonics. Asterisks: significant response amplitude from zero at $p_{FDR}<0.05$. (**B**) *Left:* spatial distribution of visual responses at time window 145–155 ms. *Right:* Mean visual responses over two ROIs in the time domain. Waveforms are shown for a time window of 233 ms during which one image is shown. *Shaded area:* standard error of the mean across participants. Blank line at around y=−1.5: stimulus onset duration. To define time windows in which amplitudes were significantly different from zero, we used a cluster-based nonparametric permutation t-test (1000 permutations, with a threshold of $p<0.05$, two-tailed) on the post-stimulus onset time points (0–1167 ms) (*Appelbaum et al., 2006*; *Blair and Karniski, 1993*).

## Visual category responses

As adults have mature category-selective regions, we next tested if using our SSVEP paradigm and the same amount of data as in infants, we identify significant category-selective responses in adults. First, we analyzed the mean response at the category frequency and its harmonics in lateral occipitotemporal ROIs. This is a selective response as it reflects the relative response to the target category (generalized across exemplars) above the general visual response to images of other categories. Despite lower response amplitudes in adults than infants, using the same amount of data as infants, adults show significant category-selective responses to each of these five categories (*Appendix 1—figure 2A* for all categories). *Appendix 1—figure 2A* shows the group-averaged Fourier amplitude spectrum over LOT and ROT ROIs (top panels) and the three-D topographies of group-averaged responses at 0.857 Hz and its first harmonic (bottom panel). We used Hotelling's T2 statistic to test the amplitude significance and found significant above-zero category-selective responses in adults at 0.857 Hz and its multiple harmonics over bilateral occipitotemporal ROIs containing seven electrodes each (ps<0.05, FDR corrected over eight levels).

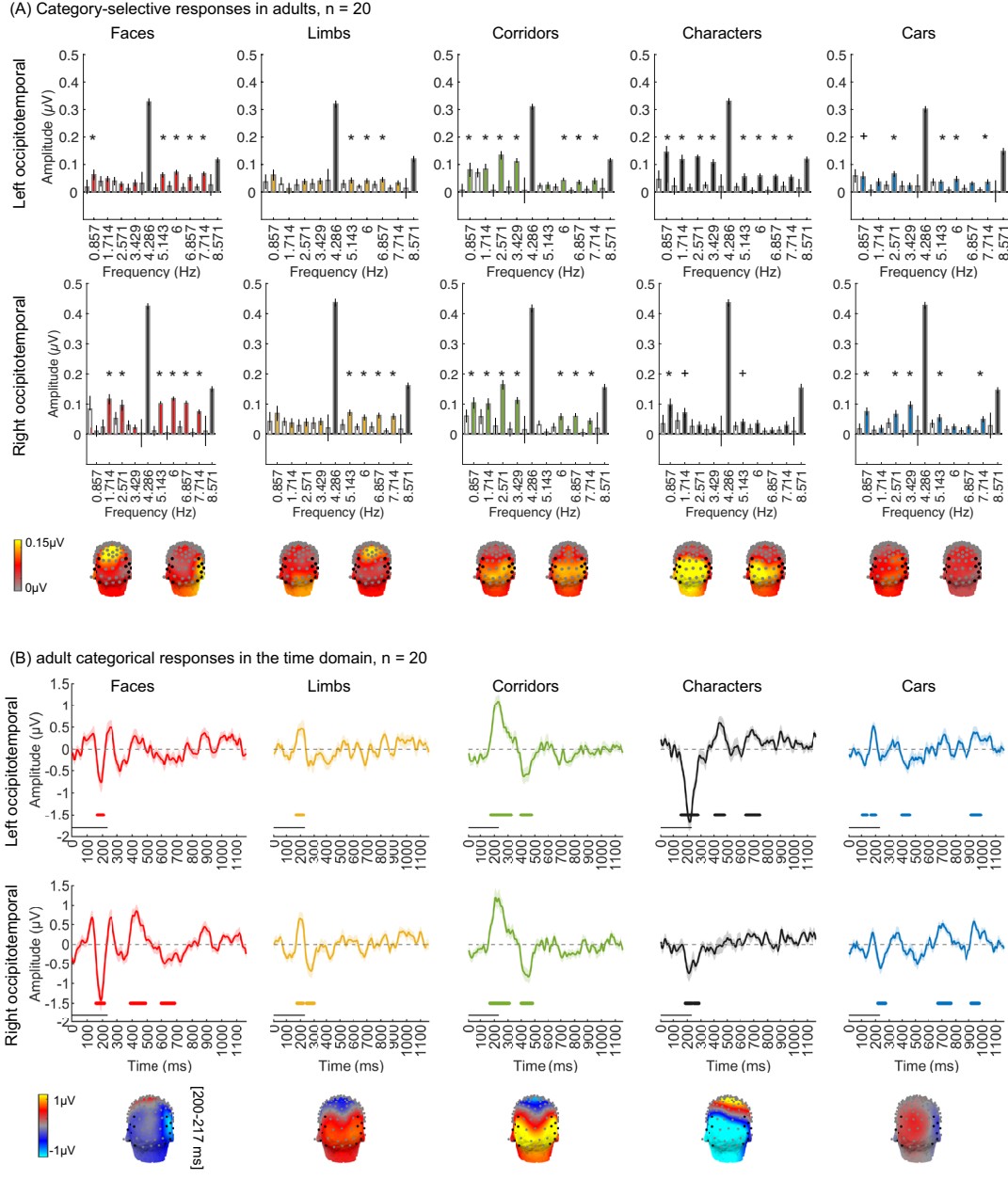

**Appendix 1—figure 2.** Adult control: using the same amount of data as infants reveals strong category-selective responses in adults' occipitotemporal cortex. (**A**) Mean Fourier amplitude spectrum across seven (left OT: 57, 58, 59, 63, 64, 65, 68; right OT: 90, 91, 94, 95, 96, 99) electrodes in bilateral occipitotemporal regions of interest (ROIs). Data are first averaged in each participant and then across 20 participants. *Error bars:* standard error of the mean across participants. *Black bars:* visual response at image frequency and harmonics; *Colored bars:* categorical response at category frequency and harmonics. *Asterisks:* significant response amplitude from zero at $p_{FDR} < 0.05$ for category harmonics. Crosses: significant response amplitude from zero at $p < 0.05$ with no FDR correction. *Black dots:* ROI channels used in analysis. (**B**) Mean category-selective responses in the time domain. Data are averaged across electrodes of each of the left and right occipitotemporal ROI in each participant and then across participants. *Colored lines along x-axis at $y = -1.5$:* significant deflections against zero (calculated with a cluster-based method, see Methods part). *Black line above x-axis:* stimulus onset duration. *Bottom panel:* spatial distribution of category-selective responses at time window 200–217 ms.

By transforming the filtered data at the category frequency and its harmonics to examine the temporal waveform of each category. Notably, each category generates a unique topography (bottom panels) at 200–217 ms after stimulus onset (*Appendix 1—figure 2B*). For faces, we found an

early negative deflection peaking ~200 ms after stimulus onset in both the LOT and ROT ROIs, with the ROT ROI showing a numerically larger mean response amplitude than the left (*Appendix 1—figure 2B*, faces). Similarly, we found an early negative peak at around 200 ms for characters, and a left hemisphere dominance (at the 162–248 ms and 398–474 ms time windows *Appendix 1—figure 2B*, characters). For both limbs and corridors, there was an early positive waveform peaking at around 200 ms in both hemispheres with no hemispheric differences.

Second, we examined in the same 20 adults whether the spatiotemporal pattern of brain responses evoked by different visual categories are distinct from one another and are reliable across participants, using an LOOCV classifier approach with spatiotemporal time series concatenated with mean temporal waveforms from three ROIs: (*Appendix 1—figure 3A*, left). Mean LOOCV classification performance was around 80% and significantly above chance in adults (*Appendix 1—figure 3A*, right). This classification was associated with correct decoding of all five categories from distributed responses in the majority of participants.

Third, we examined in the same 20 adults if there is reliable category information in spatiotemporal distributed responses in each individual. Distributed spatiotemporal responses were measured in individual participants over 23 occipital and lateral occipital ROIs using split half of data. We computed the RSM across split-halves and then calculated category distinctiveness by subtracting mean between category similarity from within-category similarity for each category. Mean RSM across 20 adults shows that spatiotemporal patterns are more similar across items of a category than across items of different categories (*Appendix 1—figure 3B*, left), and the category distinctiveness scores are significantly above zero for all five categories: faces, $t_{(19)} = 11.18$, $p_{FDR}<0.05$; limbs, $t_{(19)} = 5.01$, $p_{FDR}<0.05$; corridors, $t_{(19)} = 5.58$, $p_{FDR}<0.05$; characters, $t_{(19)} = 6.56$, $p_{FDR}<0.05$; and cars, $t_{(19)} = 10.59$, $p_{FDR}<0.05$.

Together, these analyses suggest that the experimental paradigm has sufficient power to identify category representations both at the group and individual levels.

(A) Leave-one-out-cross-validated category classifier

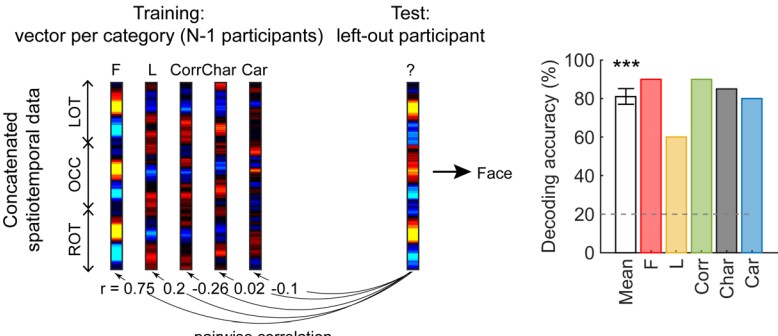

(B) Individual adult split-half analysis of distributed responses

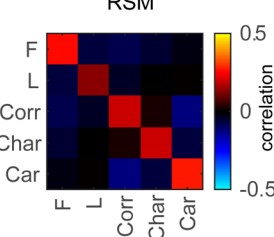

**Appendix 1—figure 3.** Adult control: using the same amount of data as infants reveals distributed category-selective responses in adults' occipitotemporal cortex. (**A**) *Left:* An illustration of winner-takes-all leave-one-out-cross-validation (LOOCV) classifier using the spatiotemporal response patterns of each category. Spatiotemporal patterns of response for each category are generated by concatenating the mean time courses from each of the three regions of interest (ROIs): left occipitotemporal (LOT), occipital (OCC), and right occipitotemporal (ROT). At each iteration, we train the classifier with the mean spatiotemporal patterns of each category from N–1 participants, and test how well it predicts the category the left-out participant is viewing from their spatiotemporal brain response. This is a winner-take-all classifier which predicts the category based on the highest pairwise

*Appendix 1—figure 3 continued on next page*

*Appendix 1—figure 3 continued*

correlation between the training and testing vectors. *Right: White:* mean decoding accuracy across all five categories. In adults, this is significantly above chance level ($t_{(19)}$ = 15.4, p<0.001). *Colored:* decoding accuracy per category. (**B**) *Left:* average adult representation similarity matrix (RSM) for odd/even splits of spatiotemporal patterns of categorical over 23 electrodes in the LOT, OCC, ROT. RSMs were generated in each participant and then averaged across all participants. *Diagonal:* correlation of distributed responses across different exemplars of the same category. *Off-diagonal:* correlations across different exemplars from different categories. *Acronyms:* F: faces; L: limbs; Corr: corridors; Char: characters; Car: Cars.

## Validation of experimental paradigm

As visual acuity develops during the first year of life (*Norcia and Tyler, 1985*; *Norcia et al., 1990*; *Appendix 1—figure 4*), one concern is that our controlled natural, gray-level stimuli may not be distinguishable to infants. Measurements of visual evoked potentials (*Norcia and Tyler, 1985*; *Norcia et al., 1990*) suggest that visual acuity in 3-month-olds is around 5–8 cycles per degree (cpd) and in 6-month-olds around 10–16 cpd (*Appendix 1—figure 4*). Thus, to simulate how our images may appear to infants, we filtered all images at 5 cpd. Despite being blurry, images of different categories are distinguishable and individual items retain their identity by visualization (Appendix 1 – *Videos 1–5*).

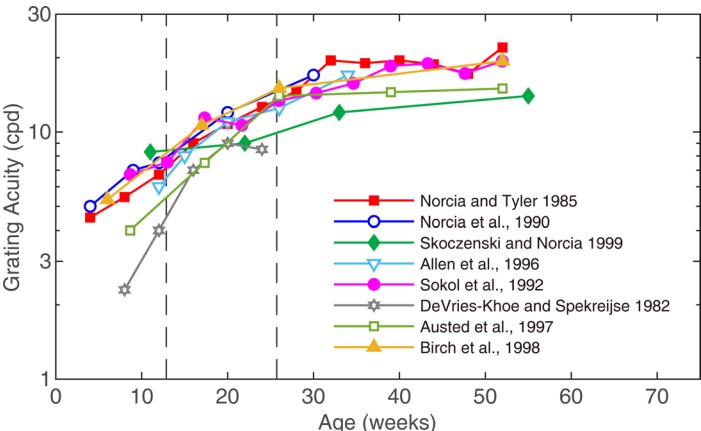

**Appendix 1—figure 4.** Grating acuity as a function of age measured with a swept spatial frequency technique combining electroencephalography (EEG). (**A**) Acuity growth functions are similar across studies, with acuity increasing from 5 to 8 cycles per degree (cpd) in 3-month-olds to around 10–16 cpd in 6-month-olds. This figure is adapted from *Norcia, 2011*.

## Visual responses over occipital cortex per condition for all age groups

In the main analysis, we averaged the image-update visual responses across five conditions for each infant, as the same visual stimuli from all five stimuli categories were viewed by the infant. However, we are showing the mean Fourier amplitude spectrum over the occipital cortex for each condition for all age groups (*Appendix 1—figure 5*). The response patterns across conditions at each age group is similar. To examine whether there are visual response amplitude differences between conditions by age groups, we quantified the RMS amplitude value of the responses at the image-update frequency (4.286 Hz) and its first three harmonics (8.571 Hz, 12.857 Hz, and 17.143 Hz) for each category condition and infant. Then, we used an LMM to test for an age by category interaction. The LMM was conducted over the posterior occipital ROI. Results of this analysis find no significant main effect of category ($\beta_{category}$ = 0.08, 95% CI: –0.08 to 0.24, $t_{(301)}$ = 0.97, p=0.33) or category by age interaction ($\beta_{category \times age}$ = –0.04, 95% CI: –0.11 to 0.03, $t_{(301)}$ = –1.09, p=0.28), which means that the visual response amplitudes are consistent across category conditions.

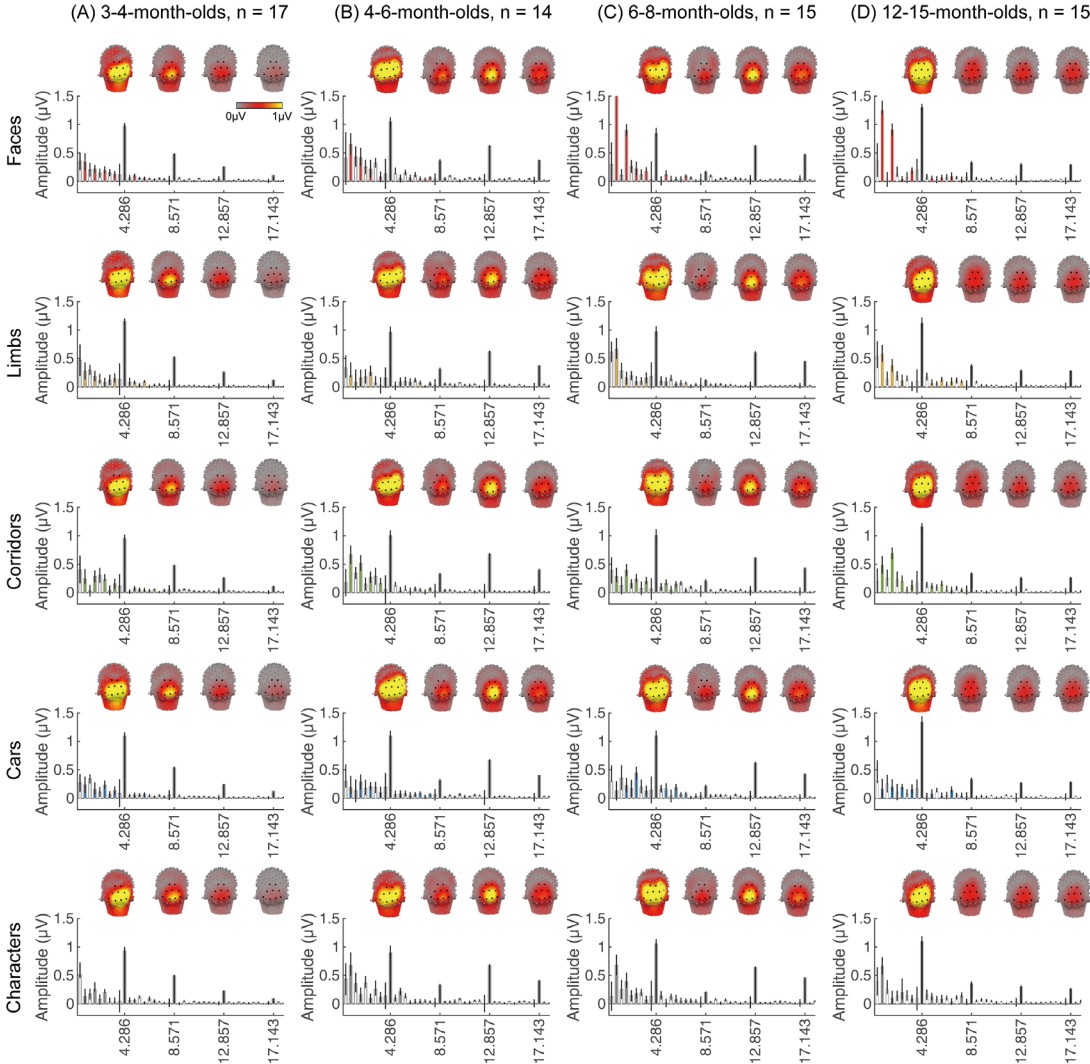

**Appendix 1—figure 5.** Visual responses over occipital cortex at the image-update frequency and its harmonics in five category conditions in all age groups. Each column shows mean responses across infants in an age group for each condition. (**A**) 3- to 4-month-olds, n=17; (**B**) 4- to 6-month-olds, n=14; (**C**) 6- to 8-month-olds, n=15; (**D**) 12- to 15-month-olds, n=15. Graphs show the mean Fourier amplitude spectrum over the occipital region of interest (ROI). The visual response is at the image-update frequency (4.286 Hz) and its first three harmonics, with the mean topographies at these frequencies of interest shown on top.

## LMM analyses of visual responses (associated with *Figure 2*)

**Appendix 1—table 4.** Peak latency of visual responses by age and time window (window 1: 60–90 ms; window 2: 90–160 ms for 3- to 4-month-olds, and 90–110 ms for other groups).
*Formula:* Peak latency ~1 + log10(age) × time window + (1|participant). Significant effects are indicated by asterisks.

| Parameter | β | CI | df | t | p |
|---|---|---|---|---|---|
| Intercept | −54.61 | −100.82, −8.41 | 118 | −2.34 | 0.021* |
| Age | 39.77 | 19.51, 60.02 | 118 | 3.89 | 0.00017*** |
| Window | 141.68 | 112.92, 170.45 | 118 | 9.75 | 7.69e-17*** |
| Age×window | −45.78 | −58.39, −33.17 | 118 | −7.19 | 6.39e-11** |

**Appendix 1—table 5.** Peak latency of visual responses by age at each of the two time windows. *Formula:* Peak latency ~1 + log10(age) + (1|participant). Significant effects are indicated by asterisks.

| Window | Parameter | β | CI | df | t | p |
|---|---|---|---|---|---|---|
| Window1 | Intercept | 90.19 | 75.66, 104.72 | 59 | 12.42 | 4.22e-18*** |
|  | Age | −7.44 | −13.82, −1.06 | 59 | −2.33 | 0.02* |
| Window2 | Intercept | 218.09 | 196.05, 240.13 | 59 | 19.80 | 9.64e-28*** |
|  | Age | −46.91 | −56.56, −37.27 | 59 | −9.73 | 7.06e-14*** |

**Appendix 1—table 6.** Analysis of peak amplitude of visual responses by age and time window. *Formula:* Peak amplitude ~1 + log10(age) × time window + (1|participant). Significant effects are indicated by an asterisk.

| Parameter | β | CI | df | t | p |
|---|---|---|---|---|---|
| Intercept | −18.69 | −32.35, −5.03 | 118 | −2.71 | 0.008** |
| Age | 3.93 | −2.05, 9.92 | 118 | 1.30 | 0.20 |
| Window | 17.24 | 8.66, 25.82 | 118 | 3.98 | 0.0001*** |
| Age×window | −4.90 | −8.66, −1.14 | 118 | −2.58 | 0.011* |

**Appendix 1—table 7.** Peak amplitude of visual responses by age at each of the two time windows. *Formula:* Peak amplitude ~1 + log10(age) + (1|participant). Significant effects are indicated by asterisks.

| Window | Parameter | β | CI | df | t | p |
|---|---|---|---|---|---|---|
| Window1 | Intercept | 1.69 | −3.42, 6.79 | 59 | 0.66 | 0.51 |
|  | Age | 0.91 | −1.33, 3.15 | 59 | 0.82 | 0.42 |
| Window2 | Intercept | 11.16 | 4.80, 17.51 | 59 | 3.51 | 0.0009*** |
|  | Age | −3.59 | −6.38, −0.81 | 59 | −2.58 | 0.012* |

## Frequency domain analyses of infants' categorical responses to limbs, corridors, characters, and cars (associated with *Figure 3*)

*Figure 3* shows the group-averaged categorical response to faces. *Appendix 1—figures 6–9* show group averaged categorical responses to the rest four conditions other than faces in four age groups. We found significant responses in 6- to 8-month-olds for limbs, corridors, and characters, and in 12–15 months for corridors and characters.

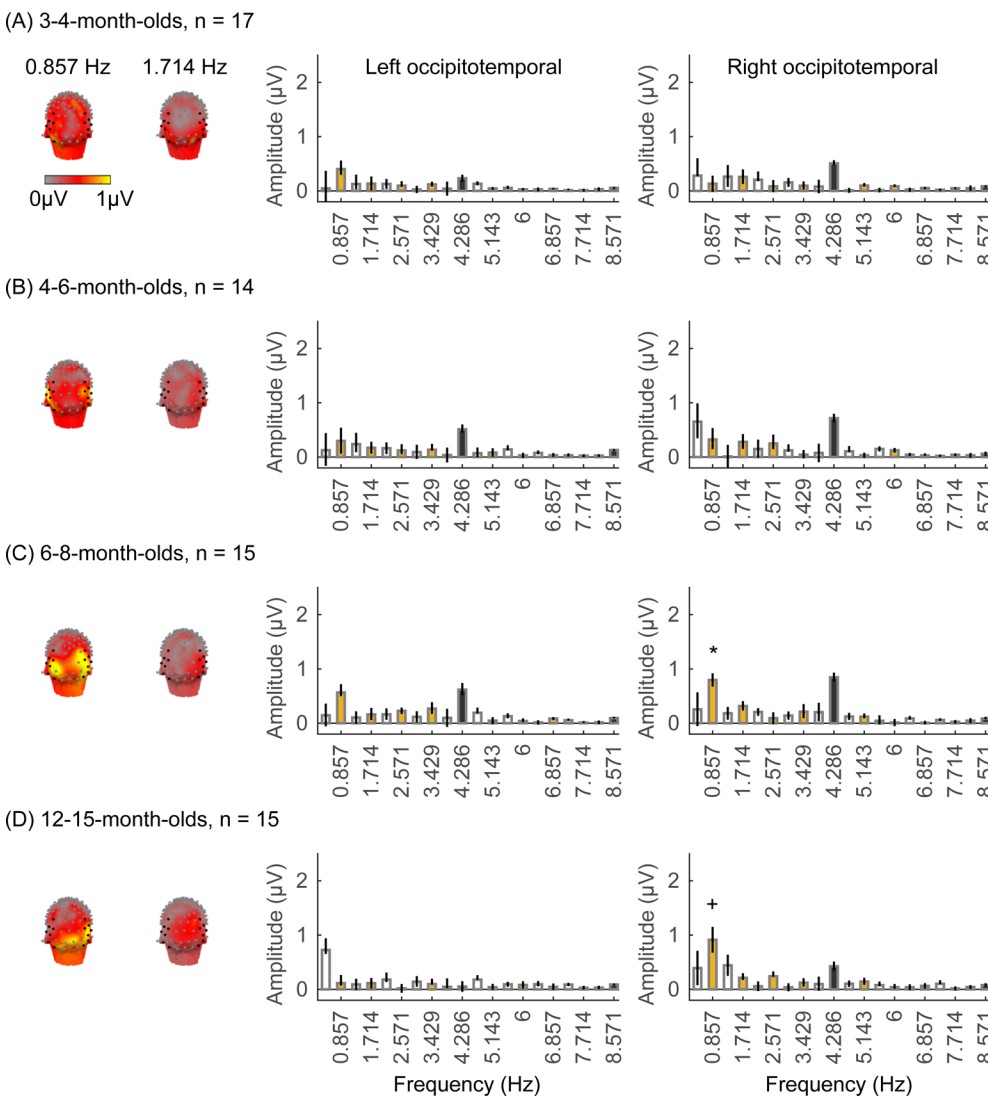

**Appendix 1—figure 6.** Limb responses emerge over occipitotemporal electrodes after 6 months of age. Each panel shows mean responses at the category frequency (0.857 Hz) and its harmonics across infants in an age group. (**A**) 3- to 4-month-olds, n=17; (**B**) 4- to 6-month-olds; n=14; (**C**) 6- to 8-month-olds, n=15; (**D**) 12- to 15-month-olds, n=15. *Left panels in each row*: spatial distribution of categorical response at 0.857 Hz and its first harmonic. Harmonic frequencies are indicated on the top. *Right two panels in each row*: mean Fourier amplitude spectrum across seven left occipitotemporal electrodes and seven right occipitotemporal (shown in black on the left panel). Data are first averaged in each participant and then across participants. *Error bars*: standard error of the mean across participants in an age group. *Asterisk*: significant amplitude vs. zero (p<0.05, FDR corrected). Cross: significant amplitude vs. zero (p<0.05, with no FDR correction). *Black bars*: image frequency and harmonics; *colored bars*: category frequency and harmonics.

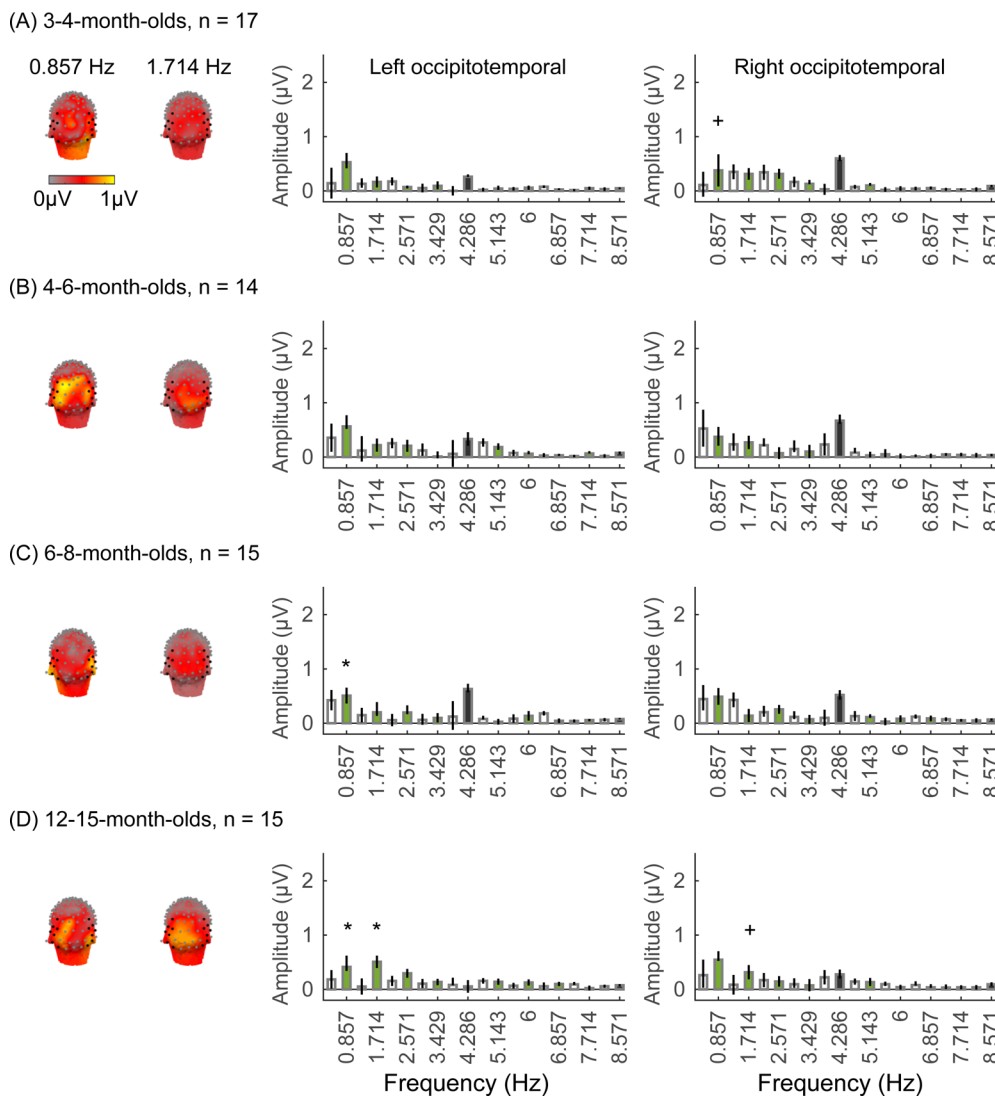

**Appendix 1—figure 7.** Corridor responses emerge over occipitotemporal electrodes after 6 months of age. Each panel shows mean responses at the category frequency and its harmonics across infants in an age group. (**A**) 3- to 4-month-olds, n=17; (**B**) 4- to 6-month-olds; n=14; (**C**) 6- to 8-month-olds, n=15; (**D**) 12- to 15-month-olds, n=15. *Left panels in each row:* spatial distribution of categorical response at 0.857 Hz and its first harmonic. Harmonic frequencies are indicated on the top. *Right two panels in each row*: mean Fourier amplitude spectrum across seven left occipitotemporal electrodes and seven right occipitotemporal (shown in black on the left panel). Data are first averaged in each participant and then across participants. *Error bars:* standard error of the mean across participants in an age group. *Asterisks:* significant amplitude vs. zero (p<0.05, FDR corrected). Crosses: significant amplitude vs. zero (p<0.05, with no FDR correction). *Black bars: image frequency and harmonics; colored bars: category frequency and harmonics.*

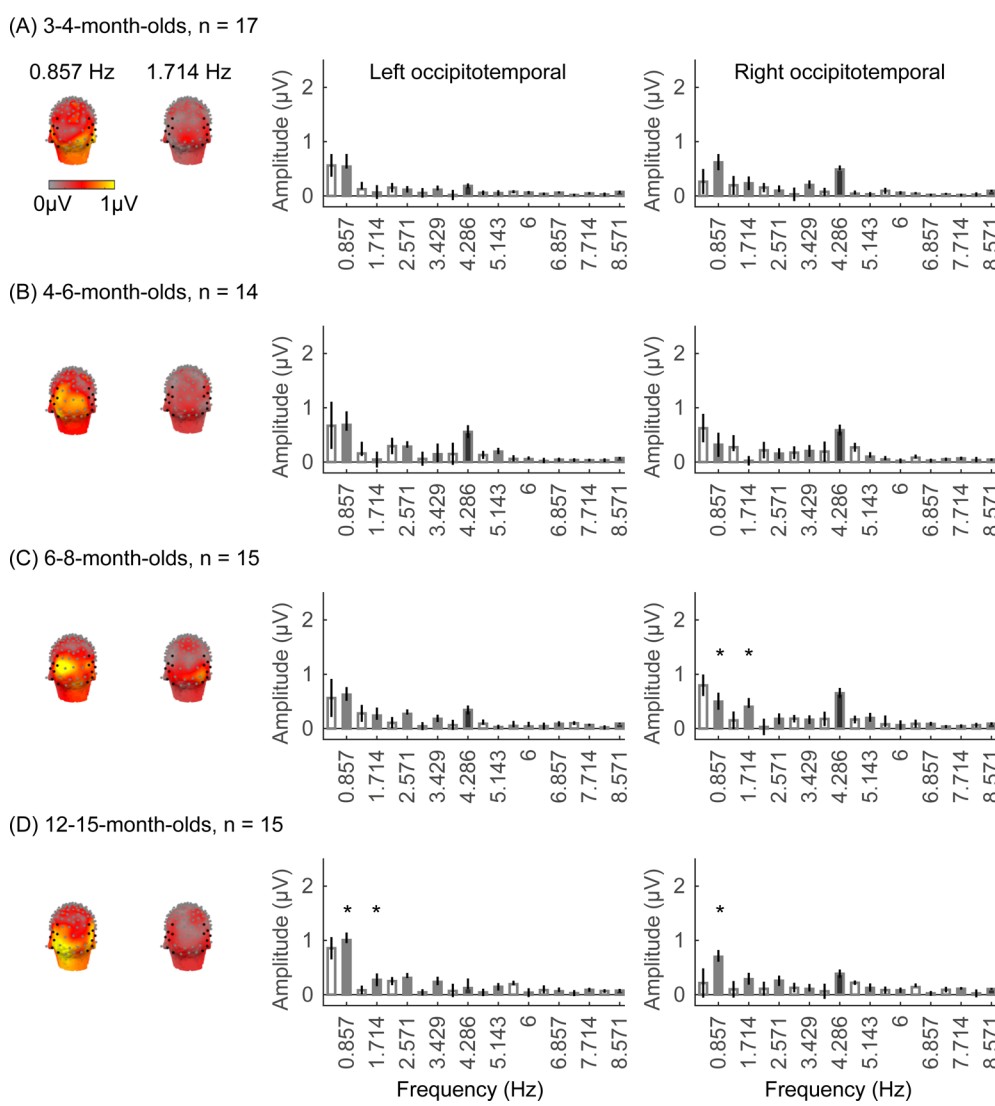

**Appendix 1—figure 8.** Significant character responses found over occipitotemporal electrodes at 12–15 months of age. Each panel shows mean responses at the category frequency (0.857 Hz) and its harmonics across infants in an age group. (**A**) 3- to 4-month-olds, n=17; (**B**) 4- to 6-month-olds; n=14; (**C**) 6- to 8-month-olds, n=15; (**D**) 12- to 15-month-olds, n=15. *Left panels in each row*: spatial distribution of categorical response at 0.857 Hz and its first harmonic. Harmonic frequencies are indicated on the top. *Right two panels in each row*: mean Fourier amplitude spectrum across seven left occipitotemporal electrodes and seven right occipitotemporal (shown in black on the left panel). Data are first averaged in each participant and then across participants. *Error bars*: standard error of the mean across participants in an age group. *Asterisks*: significant amplitude vs. zero (p<0.05, FDR corrected). *Black bars*: image frequency and harmonics; *colored bars*: category frequency and harmonics.

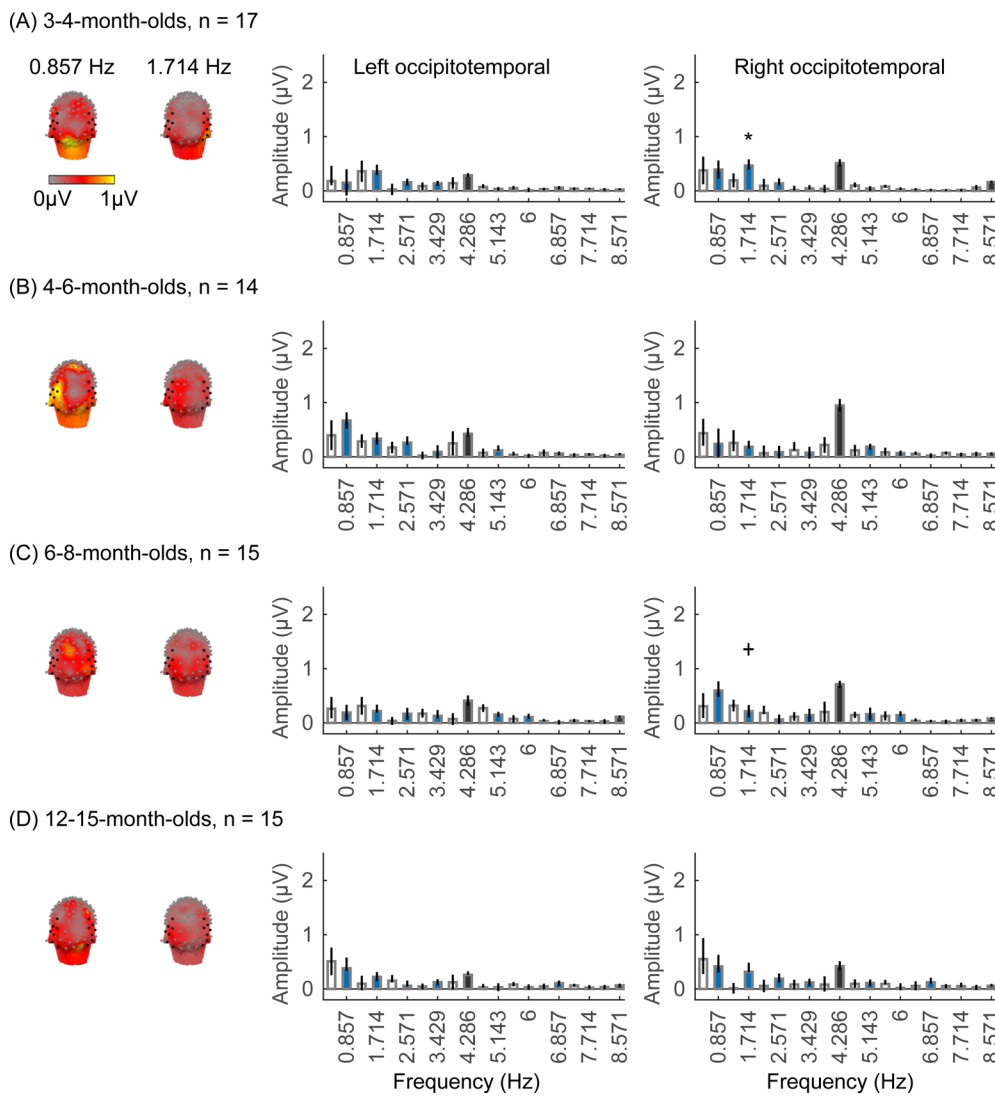

**Appendix 1—figure 9.** Significant car responses found over occipitotemporal electrodes at 3–4 months of age. Each panel shows mean responses at the category frequency and its harmonics across infants in an age group. (**A**) 3- to 4-month-olds, n=17; (**B**) 4- to 6-month-olds; n=14; (**C**) 6- to 8-month-olds, n=15; (**D**) 12- to 15-month-olds, n=15. *Left panels in each row:* spatial distribution of categorical response at 0.857 Hz and its first harmonic. Harmonic frequencies are indicated on the top. *Right two panels in each row:* mean Fourier amplitude spectrum across seven left occipitotemporal electrodes and seven right occipitotemporal (shown in black on the left panel). Data are first averaged in each participant and then across participants. *Error bars:* standard error of the mean across participants in an age group. *Asterisk:* significant amplitude vs. zero (p<0.05, FDR corrected). Cross: significant amplitude vs. zero (p<0.05, with no FDR correction). *Black bars:* image frequency and harmonics; *colored bars:* category frequency and harmonics.

## LMM analyses of category-selective responses

**Appendix 1—table 8.** Analysis of peak amplitude of waveforms of category responses by age and category.

Separate linear mixed models (LMMs) were done separately for the left occipitotemporal (OT) and right OT regions of interest (ROIs). *Formula:* Peak amplitude ~1 + log10(age) × category + (1|participant); Peak latency ~1 + log10(age) × category + (1|participant). Significant effects are indicated by an asterisk.

| ROI/metric | Parameter | β | CI | df | t | p |
|---|---|---|---|---|---|---|
| Left OT/ | Intercept | −1.35 | −7.36, 4.66 | 301 | −0.44 | 0.66 |
| amplitude | Age | 2.62 | −0.02, 5.25 | 301 | 1.95 | 0.052 |
|  | Category | 0.18 | −1.58, 1.93 | 301 | 0.20 | 0.84 |
|  | Age×category | −0.20 | −0.97, 0.57 | 301 | −0.51 | 0.61 |
| Left OT/ | Intercept | 730.29 | 477.96, 982.61 | 301 | 5.70 | 2.9e-8*** |
| latency | Age | −97.17 | −207.76, 13.43 | 301 | −1.73 | 0.08 |
|  | Category | −43.35 | −119.43, 32.73 | 301 | −1.12 | 0.26 |
|  | Age×category | 20.24 | −13.11, 53.58 | 301 | 1.19 | 0.23 |
| Right OT/ | Intercept | −7.39 | −14.44, −0.36 | 301 | −2.07 | 0.04* |
| amplitude | Age | 5.53 | 2.45, 8.62 | 301 | 3.53 | 0.0005*** |
|  | Category | 2.19 | 0.06, 4.3 | 301 | 2.02 | 0.04* |
|  | Age×category | −1.09 | −2.00, −0.14 | 301 | −2.26 | 0.02* |
| Right OT/ | Intercept | 922.47 | 667.95, 1177 | 301 | 7.13 | 7.38e-12*** |
| latency | Age | −173.17 | −284.73, −61.61 | 301 | −3.05 | 0.002** |
|  | Category | −64.41 | −141.15, 12.33 | 301 | −1.65 | 0.10 |
|  | Age×category | 28.49 | −5.15, 62.12 | 301 | 1.67 | 0.10 |

**Appendix 1—table 9.** Analysis of peak amplitude of waveforms of category responses for each category in the right occipitotemporal (OT) region of interest (ROI).

*Formula:* Peak amplitude ~age + (1|participant). Significant effects are indicated by an asterisk.

| Category | Parameter | β | CI | df | t | p |
|---|---|---|---|---|---|---|
| Faces | Intercept | −10.43 | −17.81, −3.05 | 59 | −2.83 | 0.006** |
|  | Age | 7.27 | 4.03, 10.51 | 59 | 4.50 | 3.30e-5*** |
| Limbs | Intercept | 2.10 | −3.62, 7.83 | 59 | 0.74 | 0.46 |
|  | Age | −2.90 | −5.41,−0.38 | 59 | −2.31 | 0.02* |
| Corridors | Intercept | −4.65 | −11.09, 1.81 | 59 | −1.44 | 0.15 |
|  | Age | 0.35 | −2.47, 3.18 | 59 | 0.25 | 0.80 |
| Characters | Intercept | −2.79 | −8.06, 2.49 | 59 | −1.06 | 0.3 |
|  | Age | −0.66 | −2.97, 1.65 | 59 | −0.57 | 0.57 |
| Cars | Intercept | −4.87 | −11.46, 1.73 | 59 | −1.48 | 0.15 |
|  | Age | 0.78 | −2.11, 3.67 | 59 | 0.54 | 0.59 |

## Individual-level decoding analysis

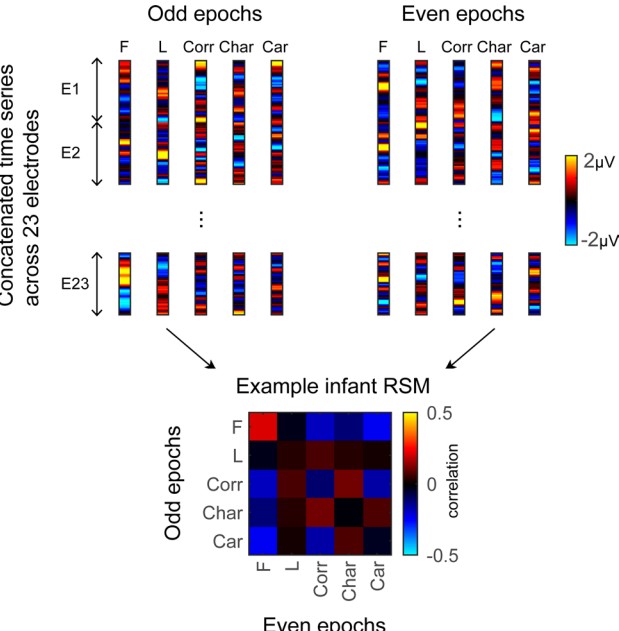

**Appendix 1—figure 10.** Illustration of the winner-takes-all (WTA) classifier. In each individual, the time series data is split into odd and even trials. We concatenate the time series data from 23 electrodes in the left occipitotemporal, occipital, and right occipitotemporal regions of interest (ROIs) into a pattern vector for each split half and each condition. The classifier is trained on one half of the data (i.e. odd or even trials) and tested on how well it could predict the rest half of the data (i.e. even or odd trials) for each individual. The bottom shows the representation similarity matrix (RSM) in an example infant. Each cell indicates the correlation between distributed responses to different images of the same (on-diagonal) or different (off-diagonal) categories. *F: faces; L: limbs; Corr: corridors; Char: characters; Car: cars.*

