## [Editor Report · eLife Assessment]

This **valuable** study investigates the development of high-level visual responses in infants, finding that neural responses specific to faces are present by 4-6 months but not earlier. The study is methodologically **convincing**, using state-of-the-art experimental design and analysis approaches. The findings would be of broad interest to the cognitive neuroscience and developmental psychology research communities.

---

## [Referee Report · Reviewer #1 (Public review)]

Summary:

In the paper, Yan and her colleagues investigate at which stage of development different categorical signals can be detected with EEG using a Steady-state visual evoked potential paradigm. The study reports the development trajectory of selective responses to five categories (i.e., faces, limbs, corridors, characters, and cars) over the first 1.5 years of life. It reveals that while responses to faces show significant early development, responses to other categories (i.e., characters and limbs) develop more gradually and emerge later in infancy. The insights the study provides are important. The paper is well-written and enjoyable, and the content is well-motivated and solid.

Strengths:

(1) This study contains a rich dataset with a good amount of effort. It covers a large sample of infants across ages (N=45) asking an interesting question about when we can robustly detect visual category representations during the first year of life of human infants.

(2) The chosen category stimuli are appropriate and well-controlled. These categories are classic and important for situating the study in the field within a well-established theoretical framework.

(3) The brain measurements are solid. Visual periodicity allows for the dissociation of selective responses to image categories within the same rapid image stream, which appears at different intervals. This is important for the infant field, where brain measures often lack sensitivity due to the developing brain's low signal-to-noise ratio and short recording time. Considering the significant changes in the brain during infancy, this robust measure of ERPs has good interpretability.

Weaknesses:

(1) There is limited data available for each category per infant, with an average of only 5 trials/epochs per category per participant. This insufficient data for each individual weakens the study, as it limits the power of analysis and constrains our understanding of the research question. If more data were available for each tested category per individual, the findings would be more robust and our ability to answer the questions more effectively would be enhanced.

(2) The study would benefit from a more detailed explanation of analysis choices, limitations, and broader interpretations of the findings. This should include: (a) improving the treatment of bias from specific categories (e.g., faces) towards others; (b) justifying the specific experimental and data analysis choices; and (c) expanding the interpretation and discussion of the results. I believe that giving more attention to these aspects would improve the study and contribute positively to the field.

Comments on revised submission:

The authors thoroughly addressed my concerns, and I have no further issues with their response.

---

## [Referee Report · Reviewer #2 (Public review)]

Summary:

The current work investigates the neural signature of category representation in infancy. Neural responses during steady-state visually-evoked potentials (ssVEPs) were recorded in four age groups of infants between 3 and 15 months. Stimuli (i.e., faces, limbs, corridors, characters, and cars) were presented at 4.286 Hz with category changes occurring at a frequency of 0.857 Hz. Results of the category frequency analyses showed that reliable responses to faces emerge around 4-6 months, whereas response to libs, corridors, and characters emerge around 6-8 months. Additionally, the authors trained a classifier for each category to assess how consistent the responses were across participants (leave-one-out approach). Spatiotemporal responses to faces were more consistent than the responses to the remaining categories and increased with increasing age. Faces showed an advantage over other categories in two additional measures (i.e., representation similarity and distinctiveness). Together, these results suggest a different developmental timing of category representation.

Strengths:

The study design is well organized. The authors described and performed analyses on several measures of neural categorization, including innovative approaches to assess the organization of neural responses. Results are in support of one of the two main hypotheses on the development of category representation described in the introduction. Specifically, the results suggest a different timing in the formation of category representations, with earlier and more robust responses emerging for faces over the remaining categories. Graphic representations and figures are very useful when reading the results. The inclusion of the adult sample and results further validate the approach utilized with infants.

Comments on revised submission:

The revised manuscript satisfactorily addressed all my previous comments.

---

## [Referee Report · Reviewer #3 (Public review)]

Yan et al. ("When do visual category representations emerge in infant brains?") present an EEG study of category-specific visual responses in infancy from 3 to 15 months of age. In their experiment, infants viewed visually controlled images of faces and several non-face categories in a steady state evoked potential paradigm. The authors find visual responses at all ages, but face responses only at 4-6 months and older, and other category-selective responses at later ages. They find that spatiotemporal patterns of response can discriminate faces from other categories at later ages.

Overall, I found the study well-executed and a useful contribution to the literature. The study advances prior work by using well-controlled stimuli, subgroups at different ages, and new analytic approaches. The data and analyses support their conclusions regarding developmental change in neural responses to high-level visual stimuli.

---

## [Author Response]

The following is the authors’ response to the original reviews.

**Public Reviews:**

**Reviewer #1 (Public Review):**
Summary:In the paper, Yan and her colleagues investigate at which stage of development different categorical signals can be detected with EEG using a steady-state visual evoked potential paradigm. The study reports the development trajectory of selective responses to five categories (i.e., faces, limbs, corridors, characters, and cars) over the first 1.5 years of life. It reveals that while responses to faces show significant early development, responses to other categories (i.e., characters and limbs) develop more gradually and emerge later in infancy. The paper is well-written and enjoyable, and the content is well-motivated and solid.Strengths:(1) This study contains a rich dataset with a substantial amount of effort. It covers a large sample of infants across ages (N=45) and asks an interesting question about when visual category representations emerge during the first year of life.(2) The chosen category stimuli are appropriate and well-controlled. These categories are classic and important for situating the study within a well-established theoretical framework.(3) The brain measurements are solid. Visual periodicity allows for the dissociation of selective responses to image categories within the same rapid image stream, which appears at different intervals. This is important for the infant field, as it provides a robust measure of ERPs with good interpretability.Weaknesses:The study would benefit from a more detailed explanation of analysis choices, limitations, and broader interpretations of the findings. This includes:a) improving the treatment of bias from specific categories (e.g., faces) towards others;b) justifying the specific experimental and data analysis choices;c) expanding the interpretation and discussion of the results.I believe that giving more attention to these aspects would improve the study and contribute positively to the field.

We thank the reviewer for their clear summary of the work and their constructive feedback. To address the reviewer’s concerns, in the revised manuscript we now provide a detailed explanation of analysis choices, limitations, and broader interpretations, as summarized in the point-by-point responses in the section: Reviewer #1 (Recommendations For The Authors) below, for which we give here an overview in points (a), (b), and (c):

(a) The reviewer is concerned that using face stimuli as one of the comparison categories may hinder the detection of selective responses to other categories like limbs. Unfortunately, because of the frequency tagging design of our study we cannot compare the responses to one category vs. only some of the other categories (e.g. limbs vs objects but not faces). In other words, our experimental design does not enable us to do this analysis suggested by the reviewer. Nonetheless, we underscore that faces compromise only ¼ of contrast stimuli and we are able to detect significant selective responses to limbs, corridors and characters in infants after 6-8 months of age even as faces are included in the contrast and the response to faces continues to increase (see Fig 4). We discuss the reviewer’s point regarding how contrast can contribute to differences in findings in the discussion on pages 12-13, lines 344-351. Full details below in Reviewer 1: Recommendations for Authors - Frequency tagging category responses.

(b) We expanded the justification of specific experimental and data analysis choices, see details below in Reviewer 1: Recommendations for Authors ->Specific choices for experiment and data analysis.

(c) We expand the interpretation and discussion, see details below in Reviewer 1: Recommendations for Authors -> More interpretation and discussion.

**Reviewer #2 (Public Review):**
Summary:The current work investigates the neural signature of category representation in infancy. Neural responses during steady-state visually-evoked potentials (ssVEPs) were recorded in four age groups of infants between 3 and 15 months. Stimuli (i.e., faces, limbs, corridors, characters, and cars) were presented at 4.286 Hz with category changes occurring at a frequency of 0.857 Hz. The results of the category frequency analyses showed that reliable responses to faces emerge around 4-6 months, whereas responses to libs, corridors, and characters emerge at around 6-8 months. Additionally, the authors trained a classifier for each category to assess how consistent the responses were across participants (leave-one-out approach). Spatiotemporal responses to faces were more consistent than the responses to the remaining categories and increased with increasing age. Faces showed an advantage over other categories in two additional measures (i.e., representation similarity and distinctiveness). Together, these results suggest a different developmental timing of category representation.Strengths:The study design is well organized. The authors described and performed analyses on several measures of neural categorization, including innovative approaches to assess the organization of neural responses. Results are in support of one of the two main hypotheses on the development of category representation described in the introduction. Specifically, the results suggest a different timing in the formation of category representations, with earlier and more robust responses emerging for faces over the remaining categories. Graphic representations and figures are very useful when reading the results.Weaknesses:(1) The role of the adult dataset in the goal of the current work is unclear. All results are reported in the supplementary materials and minimally discussed in the main text. The unique contribution of the results of the adult samples is unclear and may be superfluous.(2) It would be useful to report the electrodes included in the analyses and how they have been selected.

We thank the reviewer for their constructive feedback and for summarizing the strengths and weaknesses of our study. We revised the manuscript to address these two weaknesses.

(1) The reviewer indicates that the role of the adult dataset is unclear. The goal of testing adult participants was to validate the EEG frequency tagging paradigm. We chose to use adults because a large body of fMRI research shows that both clustered and distributed responses to visual categories are found in adults’ high-level visual cortex. Therefore, the goal of the adult data is to determine whether with the same amount of data as we collect on average in infants, we have sufficient power to detect categorical responses using the frequency tagging experimental paradigm as we use in infants. Because this data serves as a methodological validation purpose, we believe it belongs to the supplemental data.

We clarify this in the Results, second paragraph, page 5 where now write: “As the EEG-SSVEP paradigm is novel and we are restricted in the amount of data we can obtain in infants, we first tested if we can use this paradigm and a similar amount of data to detect category-selective responses in adults. Results in adults validate the SSVEP paradigm for measuring category-selectivity: as they show that (i) category-selective responses can be reliably measured using EEG-SSVEP with the same amount of data as in infants (Supplementary Figs S1-S2), and that (ii) category information from distributed spatiotemporal response patterns can be decoded with the same amount of data as in infants (Supplementary Fig S3).”

(2) The reviewer asks us to report the electrodes used in the analysis and their selection. We note that the selection of electrodes included in the analyses has been reported in our original manuscript (Methods, section: Univariate EEG analyses). On pages 18-19, lines 530-538, we write: “Both image update and categorical EEG visual responses are reported in the frequency and time domain over three regions-of-interest (ROIs): two occipito-temporal ROIs (left occipitotemporal (LOT): channels 57, 58, 59, 63, 64, 65 and 68; right occipitotemporal (ROT) channels: 90, 91, 94, 95, 96, 99, and 100) and one occipital ROI (channels 69, 70, 71, 74, 75, 76, 82, 83 and 89). These ROIs were selected a priori based on a previously published study51. We further removed several channels in these ROIs for two reasons: (1) Three outer rim channels (i.e., 73, 81, and 88) were not included in the occipital ROI for further data analysis for both infant and adult participants because they were consistently noisy. (2) Three channels (66, 72, and 84) in the occipital ROI, one channel (50) in the LOT ROI, and one channel (101) in the ROT ROI were removed because they did not show substantial responses in the group-level analyses.”

In the section Reviewer 2, Recommendations for the authors, we also addressed the reviewer’s minor points.

**Reviewer #3 (Public Review):**
Yan et al. present an EEG study of category-specific visual responses in infancy from 3 to 15 months of age. In their experiment, infants viewed visually controlled images of faces and several non-face categories in a steady state evoked potential paradigm. The authors find visual responses at all ages, but face responses only at 4-6 months and older, and other category-selective responses at later ages. They find that spatiotemporal patterns of response can discriminate faces from other categories at later ages.Overall, I found the study well-executed and a useful contribution to the literature. The study advances prior work by using well-controlled stimuli, subgroups of different ages, and new analytic approaches.I have two main reservations about the manuscript: (1) limited statistical evidence for the category by age interaction that is emphasized in the interpretation; and (2) conclusions about the role of learning and experience in age-related change that are not strongly supported by the correlational evidence presented.

We thank the reviewer for their enthusiasm and their constructive feedback.

(1) The overall argument of the paper is that selective responses to various categories develop at different trajectories in infants, with responses to faces developing earlier. Statistically, this would be most clearly demonstrated by a category-by-age interaction effect. However, the statistical evidence for a category by interaction effect presented is relatively weak, and no interaction effect is tested for frequency domain analyses. The clearest evidence for a significant interaction comes from the spatiotemporal decoding analysis (p. 10). In the analysis of peak amplitude and latency, an age x category interaction is only found in one of four tests, and is not significant for latency or left-hemisphere amplitude (Supp Table 8). For the frequency domain effects, no test for category by age interaction is presented. The authors find that the effects of a category are significant in some age ranges and not others, but differences in significance don't imply significant differences. I would recommend adding category by age interaction analysis for the frequency domain results, and ensuring that the interpretation of the results is aligned with the presence or lack of interaction effects.

The reviewer is asking for additional evidence for age x category interaction by repeating the interaction analysis in the frequency domain. The reason we did not run this analysis in the original manuscript is that the categorical responses of interest are reflected in multiple frequency bins: the category frequency (0.857 Hz) and its harmonics, and there are arguments in the field as to how to quantify response amplitudes from multiple frequency bins (Peykarjou, 2022). Because there is no consensus in the field and also because how the different harmonics combine depends not just on their amplitudes but also on their phase, we chose to transform the categorical responses across multiple frequency bins from the frequency domain to the time domain. The transformed signal in the time domain includes both phase and amplitude information across the category frequency and its harmonics. Therefore, subsequent analyses and statistical evaluations were done in the time domain.

However, we agree with the reviewer that adding category by age interaction analysis for the frequency domain results can further solidify the results. Thus, in the revised manuscript we added a new analysis, in which we quantified the root mean square (RMS) amplitude value of the responses at the category frequency (0.857 Hz) and its first harmonic (1.714 Hz) for each category condition and infant. Then we used a LMM to test for an age by category interaction. The LMM was conducted separately for the left and right lateral occipitotemporal ROIs. Results of this analysis find a significant category by age interaction, that is, in both hemispheres, the development of response RMS amplitudes varied across category (left occipitotemporal ROIs: βcategory x age = -0.21, 95% CI: -0.39 – -0.04, *t(301)* = -2.40, *pFDR* < .05; right occipitotemporal ROIs: βcategory x age = -0.26, 95% CI: -0.48 – -0.03, *t(301)* = -2.26, *pFDR* < .05). We have added this analysis in the manuscript, pages 7-8, lines 186-193: “We next examined the development of the category-selective responses separately for the right and left lateral occipitotemporal ROIs. The response amplitude was quantified by the root mean square (RMS) amplitude value of the responses at the category frequency (0.857 Hz) and its first harmonic (1.714 Hz) for each category condition and infant. With a LMM analysis, we found significant development of response amplitudes in the both occipitotemporal ROIs which varied by category (left occipitotemporal ROIs: βcategory x age = -0.21, 95% CI: -0.39 – -0.04, t(301) = -2.40, pFDR < .05; right occipitotemporal ROIs: βcategory x age = -0.26, 95% CI – -0.48 – -0.03, t(301) = -2.26, pFDR < .05, LMM as a function of log (age) and category; participant: random effect).” We also added the formula for the LMM analysis in Table 1 in the Methods section, page 21.

(2) The authors argue that their results support the claim that category-selective visual responses require experience or learning to develop. However, the results don't bear strongly on the question of experience. Age-related changes in visual responses could result from experience or experience-independent maturational processes. Finding age-related change with a correlational measure does not favor either of these hypotheses. The results do constrain the question of experience, in that they suggest against the possibility that category-selectivity is present in the first few months of development, which would in turn suggest against a role of experience. However the results are still entirely consistent with the possibility of age effects driven by experience-independent processes. The manner in which the results constrain theories of development could be more clearly articulated in the manuscript, with care taken to avoid overly strong claims that the results demonstrate a role of experience.

Thanks for the comment. We agree with this nuanced point. It is possible that development of category-selective visual responses is a maturational process. In response to this comment, we have revised the manuscript to discuss both perspectives, see revised discussion section – A new insight about cortical development: different category representations emerge at different times during infancy, pages 14-15, lines 403-426, where we now write: “In sum, the key finding from our study is that the development of category selectivity during infancy is non-uniform: face-selective responses and representations of distributed patterns develop before representations to limbs and other categories. We hypothesize that this differential development of visual category representations may be due to differential visual experience with these categories during infancy. This hypothesis is consistent with behavioral research using head-mounted cameras that revealed that the visual input during early infancy is dense with faces, while hands become more prevalent in the visual input later in development and especially when in contact with objects 41,42. Additionally, a large body of research has suggested that young infants preferentially look at faces and face-like stimuli 17,18,33,34, as well as look longer at faces than other objects 41, indicating that not only the prevalence of faces in babies’ environments but also longer looking times may drive the early development of face representations. Further supporting the role of visual experience in the formation of category selectivity is a study that found that infant macaques that are reared without seeing faces do not develop face-selectivity but develop selectivity to other categories in their environment like body parts40. An alternative hypothesis is that differential development of category representations is maturational. For example, we found differences in the temporal dynamics of visual responses among four infant age groups, which suggests that the infant’s visual system is still developing during the first year of life. While the mechanisms underlying the maturation of the visual system in infancy are yet unknown, they may include myelination and cortical tissue maturation 66-71. Future studies can test these alternatives by examining infants’ visual diet, looking behavior, and brain development and examine responses using additional behaviorally relevant categories such as food 72–74. These measurements can test how environmental and individual differences in visual experiences may impact infants’ developmental trajectories. Specifically, a visual experience account predicts that differences in visual experience would translate into differences in development of cortical representations of categories, but a maturational account predicts that visual experience will have no impact on the development of category representations.”

**Recommendations for the authors:**

**Reviewer #1 (Recommendations For The Authors):**
Major points:Bias from faces to other categories:- Frequency tagging category responses:We see faces from non-face objects and limbs from non-limb objects. Non-limb objects include faces; I suspect that finding the effects of limbs is challenging with faces in the non-limbs category. How would you clarify the choice of categories, and to what extent are the negative (i.e., non-significant) effects on other categories not because of the heavy bias to faces?

The reviewer is concerned that using face stimuli as one of the comparison categories may hinder the ability to detect selective responses to other categories like limbs in our study. Unfortunately, because of the frequency tagging design of our study, we cannot compare the responses to one category to only some of the other categories (e.g. limbs vs objects but not faces), so our experimental design does not enable us to do the analysis suggested by the reviewer. Nonetheless, we underscore that faces compromise only ¼ of contrast stimuli in the category frequency tagging and we are able to detect significant selective responses to limbs, corridors and characters in infants after 6-8 months of age, when faces are included in the contrast and the responses to faces continue to increase more than for other categories (see Fig 4).

We address this point in the discussion where we consider differences between our findings and those of Kosakowski et al. 2022, on pages 12-13, lines 344-351 we write: “We note that, the studies differ in several ways: (i) measurement modalities (fMRI in 27 and EEG here), (ii) the types of stimuli infants viewed: in 27 infants viewed isolated, colored and moving stimuli, but in our study, infants viewed still, gray-level images on phase-scrambled backgrounds, which were controlled for several low level properties, and (iii) contrasts used to detect category-selective responses, whereby in 27 the researchers identified within predefined parcels – the top 5% of voxels that responded to the category of interest vs. objects, here we contrasted the category of interest vs. all other categories the infant viewed. Thus, future research is necessary to determine whether differences between findings are due to differences in measurement modalities, stimulus format, and data analysis choices.”

- Decoding analyses:Figure 5 Winner-take-all classification. First, the classifier may be biased towards the categories with strong and clean data, similar to the last point, this needs clarification on the negative effect. Second, it could be helpful to see how exactly the below-chance decoded categories were being falsely classified to which categories at the group level. Decoding accuracy here means a 20% chance the selection will go to the target category, but the prediction and the exact correlation coefficient the winner has is not explicit; concerning a value of 0.01 correlation could take the winner among negative or pretty bad correlations with other categories. It would be helpful to report how exactly the category was correlated, as it could be a better way to define the classification bias, for example, correlation differences between hit and miss classification. Also, the noise ceiling of the correlation within each group should be provided. Third, this classifier needs improvement in distinguishing between noise and signals to identify the type of information it extracts. Do you have thoughts about that?

Thanks for the questions, answers below:

In the winner-take-all (WTA) classifier analysis, at each iteration, the LOOCV classifier computed the correlation between each of the five category vectors from the left-out participant (test data, for an unknown stimulus) and each of the mean spatiotemporal vectors across the N-1 participants (training data, labeled data). The winner-take-all (WTA) classifier classifies the test vector to the category that yields the highest correlation with the training vector. For a given test pattern, correct classification yielded a score of 1 and an incorrect classification yielded a score of 0. Then we computed the group mean decoding performance across all N iterations for each category and the group mean decoding accuracies across five categories.

For the classification data in Fig 5, the statistics and differences from chance are provided in 5B, where we report overall classification across all categories from an infant’s brain data. Like the reviewer, we were interested in assessing if successful classification is uniform across categories or is driven by some categories. As is visible in 5C, decoding success is non-uniform across categories, and is higher for faces than other categories. Because this is broken by category we cannot compare to chance, and what is reported in Fig 5c is percentage infants in each age group that a particular category was successfully decoded. Starting from 4 months of age, faces can be decoded from distributed brain data in a majority of infants, but other categories only in 20-40% of infants.

The reviewer also asks about what levels of correlations drive the classification. The analysis of RSMs in Fig 6a shows the mean correlations of distributed responses to different images within and between categories per age group. As is evident from the RSM, reproducible responses for a category only start to emerge at 4-6 months of age and the highest within category correlations are for faces. To quantify what drives the classification we measure distinctiveness - within category minus between-category correlations of distributed responses; all individual infant data per category are in Fig 6C. Distinctiveness values vary by age and category, see text related to Fig 6 in section: What is the nature of categorical spatiotemporal patterns in individual infants?

Figure 6 Category distinctiveness. An analysis that runs on a "single item level" would ideally warrant a more informative category distinction. Did you try that? Does it work?

Thanks for the question. We agree that doing an analysis at the single item level would be interesting. However, none of the images were repeated, so we do not have sufficient SNR to perform this analysis.

Specific choices for experiment and data analysis:- Although using the SSVEP paradigm is familiar to the field, the choice could be detailed for understanding or evaluation of the effectiveness of the paradigm. For example, how the specific frequency for entrainment was chosen, and are there any theories or related warrants for studying in infants?

Thanks for the questions. We choose to use the SSVEP paradigm over traditional ERP designs for several reasons, as described which have been listed in our original manuscript (Results part, first paragraph, pages 4-5, lines 90-94): “We used the EEG-SSVEP approach because: (i) it affords a high signal-to-noise ratio with short acquisitions making it effective for infants 23,46, (ii) it has been successfully used to study responses to faces in infants23,46,49, and (iii) it enables measuring both general visual response to images by examining responses at the image presentation frequency (4.286 Hz), as well as category-selective responses by examining responses at the category frequency (0.857 Hz, Fig 1A).”

With regards to our choice of presentation rate, a previous study in 4-6-month-olds by de Heering and Rossion (2015) used SSVEP showing infants faces and objects presented the visual stimuli at 6 Hz (i.e. 167 ms per image) to study infants’ categorical responses to natural faces relative to objects. Here, we chose to use a relatively slower presentation rate, which was 4.286 Hz (i.e. 233 ms per image), so that our infant participants would have more time to process each image yet still unlikely to make eye movements across a stimulus. Both de Heering et (2015) and our study have found significant selective responses to faces relative to other categories in 4-6-month-olds, across these presentation rates. As discussed in a recent review of frequency tagging with infants: The visual oddball paradigm (Peykarjou, 2022), there are many factors to consider when adapting SSVEP paradigms to infants. We agree that an interesting direction for future studies is examination of how SSVEP parameters such as stimulus and oddball presentation rate, and overall duration of acquisition affects the sensitivity of the SSVEP paradigm in infants. We added a discussion point on this on page 12, lines 332-334 where we write: “As using SSVEP to study high-level representations is a nascent field52–54, future work can further examine how SSVEP parameters such as stimulus and target category presentation rate may affect the sensitivity of measurements in infants (see review by54).”

- There is no baseline mentioned in the study. How was the baseline considered in the paradigm and data analysis? The baseline is important for evaluating how robust/ reliable the periodic responses within each group are in the first place. It also helps us to see how different the SNR changes in the fast periodic responses from baseline across age groups. Would the results be stable if the response amplitudes were z-scored by a baseline?

Thanks for the question. Previous studies using a similar frequency tagging paradigm have compared response amplitude at stimulus-related frequencies to that of neighboring frequency bins as their baseline for differentiating signal from noise. We use a more statistically powerful method, the Hotelling’s T2 statistic to test whether response amplitudes were statistically different from 0 amplitude. Importantly, this method takes into consideration both the amplitude and phase information of the response. That is, a significant response is expected to have consistent phase information across participants as well as significant amplitude.

- Statistical inferences: could the variance of data be considered appropriately in your LLM? Why?

As we have explained in our original manuscript (Methods part, section-Statistical Analyses of Developmental Effects, page 21 lines 611-615): “LMMs allow explicit modeling of both within-subject effects (e.g., longitudinal measurements) and between-subject effects (e.g., cross-sectional data) with unequal number of points per participants, as well as examine main and interactive effects of both continuous (age) and categorical (e.g., stimulus category) variables. We used random-intercept models that allow the intercept to vary across participants (term: 1|participant).” This statistical model is widely used in developmental studies that combine both longitudinal and cross-sectional measurements (e.g. Nordt et al. 2022, 2023; Natu et al. 2021; Grotheer et al. 2022).

- The sampling of the age groups. Why are these age groups considered, as 8-12 months are not considered? Or did the study first go with an equal sampling of the ages from 3 to 15 months? Then how was the age group defined? The log scale of age makes sense for giving a simplified view of the effects, but the sampling procedure could be more detailed.

Thanks for the question. Our study recruited infants longitudinally for both anatomical MRI and EEG studies. Some of the infants participated in both studies and some only in one of the studies. Infants were recruited at around newborn, 3 months, 6 months, and 12 months. We did not recruit infants between 8-12 months of age because around 9 months there is little contrast between gray and white matter in anatomical MRI scans that were necessary for the MRI study. For the EEG study we binned the subjects by age group such that there were a similar number of participants across age groups to enable similar statistical power. The division of age groups was decided based on the distribution of the infants included in the analyses.

We have now added the sampling procedure details in the Methods, part, under section: Participants, pages 15-16, lines 440-445: “Sixty-two full-term, typically developing infants were recruited. Twelve participants were part of an ongoing longitudinal study that obtained both anatomical MRI and EEG data in infants. Some of the infants participated in both studies and some only in one of the studies. Infants were recruited at around newborn, 3 months, 6 months, and 12 months. We did not recruit infants between 8-12 months of age because around 9 months there is little contrast between gray and white matter in anatomical MRI scans that were necessary for the MRI study.”

- 30 Hz cutoff is arbitrary, but it makes sense as most EEG effects can be expected in a lower frequency band than higher. However, this specific choice is interesting and informative, when faced with developmental data and this type of paradigm. Would the results stay robust as the cutoff changes? Would the results benefit from going even lower into the frequency cutoff?

In the time domain analyses, we choose the 30 Hz cutoff to be consistent with previous EEG studies including those done with infants. However, as our results from the frequency domain (Fig 3, right panel, and supplementary Fig S6-S9) show that there are barely any selective categorical responses above about 6 Hz. Therefore, we expect that using a lower frequency cutoff, such as 10 Hz, will not lead to different results.

More interpretation and discussion:- You report the robust visual responses in occipital regions, the responses that differ across age groups, and their characteristics (i.e., peak latency and amplitude) in time curves. This part of the results needs more interpretation to help the data be better situated in the field; I wondered whether this relates to the difference in the signal processing of the information. Could this be the signature of slow recurrence connection development? Or how could this be better interpreted?

Thanks for the question. Changes in speed of processing can arise from several related reasons including (i) myelination of white matter connections that would lead to faster signal transmission (Lebenberg et al. 2019; Grotheer et al. 2022), (ii) maturation of cortical visual circuits affecting temporal integration time, and (iii) development of feedback connections. Our data cannot distinguish among these different mechanisms. Future studies that combine functional high temporal resolution measurements with structural imaging of tissue properties could elucidate changes in cortical dynamics over development.

We added this as a discussion point, on page 15 lines 416-420 we write: “For example, we found differences in the temporal dynamics of visual responses among four infant age groups, which suggests that the infant’s visual system is still developing during the first year of life. While underlying maturational mechanisms are yet unknown, they may include myelination and cortical tissue maturation68–73.”

- The supplementary material includes a detailed introduction to the methods when facing the developing visual acuity, which justifies the choice of the paradigm. I appreciate this thorough explanation. Interestingly, high visual acuity has its potential developmental downside; for instance, low visual acuity would aid in the development of holistic processing associated with face recognition (as discussed by Vogelsang et al., 2018, in PNAS). How do you view this point in relation to the emergence of complex cognitive processes, as here the category-selective responses?

Thanks for linking this to the Vogelsang (2018) study. Just as faces are processed in a hierarchical manner, starting with low-level features (edges, contours) and progressing to high-level features (identity, expression), other complex visual categories like cars, scenes, and body parts follow similar hierarchies. Early holistic processing could provide a foundation for recognizing objects quickly and efficiently, while feature-based processing might allow for more precise recognition and categorization as acuity increases. Therefore, as visual acuity improves, an infant’s brain can integrate finer details into those holistic representations, supporting more refined and complex cognitive processes. The balance between low- and high-level visual acuity highlights the intricate interplay between sensory processing and cognitive development across various domains.

Minor points:Paradigm:- Are the colored cartoon images for motivating infants' fixation counterbalanced across categories in the paradigm? Or how exactly were the cartoon images presented in the paradigm?

Response: Yes, the small cartoon images that were presented at the center of the screen during stimuli presentation were used to engage infants’ attention and accommodation to the screen. For each condition, they were randomly drawn from a pool of 70 images (23 flowers, 22 butterflies, 25 birds) from categories unrelated to the ones under test. They were presented in random order with durations uniformly distributed between 1 and 1.5 s. We have added these details of the paradigm to the Methods section, page 17, lines 479-481: “To motivate infants to fixate and look at the screen, we presented at the center of the screen small (~1°) colored cartoon images such as butterflies, flowers, and ladybugs. They were presented in random order with durations uniformly distributed between 1 and 1.5 s.”

Analysis:- Are the visual responses over the occipital cortex different across different category conditions in the first place? I guess this should not be different; this probably needs one more supplementary figure.

The visual responses reflect the responses to images that are randomly drawn from the five stimuli categories at a presentation frequency of 4.286 Hz. The only difference between the five conditions is that the stimuli presentation order is different. Therefore, the visual response over the occipital cortex across conditions should not be different within an age group.

In the revised manuscript, we have added Supplementary Figure S5 that shows the frequency spectra distribution and the response topographies of the visual response at 4.286 Hz and its first 3 harmonics separately for each condition and age group and a new Supplementary Materials section: 5. Visual responses over occipital cortex per condition for all age groups. On page 5, lines 116-120, we now write: “Analysis of visual responses in the occipital ROI separately by category condition revealed that visual responses were not significantly across category condition Supplementary Fig S5, no significant main effect of category (βcategory = 0.08, 95% CI: -0.08 – 0.24, t(301) = 0.97, p = .33), or category by age interaction (βcategory x age = -0.04, 95% CI: -0.11 – 0.03, t(301) = -1.09, p = .28, LMM on RMS of response to first three harmonics).”

- The summary of epochs used for each category for each age group needs to be included; this is important while evaluating whether the effects are due to not having enough data for categories or others.

This part of information is provided in the manuscript in the Methods section, page 18 lines 521-524, and supplementary Table S2. Our analysis shows that there was no significant difference in the number of pre-processed epochs across different age groups (*F(3,57)* = 1.5, *p* = .2).

- Numbers of channels of EEG being interpolated should be provided; is that a difference across age groups?

Thanks for the suggestion. We have now added information about the number of channels being interpolated for each age groups in the Methods section (page 18, lines 525-528): “The number of electrodes being interpolated for each age group were 10.0 ± 4.8 for 3-4-month-olds, 9.9 ± 3.7 for 4-6-month-olds, 9.9 ± 3.9 for 6-8-month-olds, and 7.7 ± 4.7 for 12-15-month-olds. There was no significant difference in the number of electrodes being interpolated across infant age-groups (F(3,55) = 0.78, p = .51).”

- I noticed that the removal of EEG artifacts (i.e., muscles and eye-blinks) for data analysis is missing; did the preprocessing pipeline involve any artifacts removing procedures that are typically used in both infants and adults SSVEP data analysis? If so, please provide more information.

In our analysis, artifact rejection was performed in two steps. First, the continuous filtered data were evaluated according to a sample-by-sample thresholding procedure to locate consistently noisy channels. Channels with more than 20% of samples exceeding a 100-150 μV amplitude threshold were replaced by the average of their six nearest spatial neighbors. Once noisy channels were interpolated in this fashion, the EEG was re-referenced from the Cz reference used during the recording to the common average of all sensors and segmented into epochs (1166.7-ms). Finally, EEG epochs that contained more than 15% of time samples exceeding threshold (150-200 microvolts) were excluded on a sensor-by-sensor basis. This method is provided in the manuscript under Methods section, page 18 lines 510-516.

Figure:- Supplementary Figure 8. The illustration of the WTA classifier was not referred to anywhere in the main text.

Thanks for pointing this out. The supplementary Figure 8 should be noted as supplementary Figure 10 instead. We have now mentioned it in the manuscript, page 10, line 267.

- Figure 5 WTA classifier needed to be clarified. It was correlation-based but used to choose the most correlated response patterns averaged across the N-1 subjects for the leave-one-out subject. The change from correlation coefficients to decoding accuracy could be clearer as I spent some time making sense of it. The correlation coefficient here evaluates how correlated the two vectors are, but the actual decoding accuracy estimated at the end is the percentage of participants who can be assigned to the "ground truth" label, so one step in between is missing. Can this be better illustrated?

Thanks for surfacing that this is not described sufficiently clearly and for your suggestions. The spatiotemporal vector was calculated separately for each category. This is illustrated in Fig 5A. At each iteration, the LOOCV classifier computed the correlation between each of the five category vectors from the left-out participant (test data, for an unknown stimulus) and each of the mean spatiotemporal vectors across the N-1 participants (training data, labeled data). The winner-take-all (WTA) classifier classifies the test vector to the category that yields the highest correlation with the training vector. This is illustrated in Fig 5A, with spatiotemporal patterns and correlation values from an example infant shown. For a given test pattern, correct classification yields a score of 1 and an incorrect classification yields a score of 0. We compute the percentage correct across all categories for each left-out-infant, and then mean decoding performance across all participants in an age group (Fig 5B). We have now added these details in the Methods part, section – Decoding analyses, Group-level, page 20 lines 590-597, where we write: “At each iteration, the LOOCV classifier computed the correlation between each of the five category vectors from the left-out participant (test data, for an unknown stimulus) and each of the mean spatiotemporal vectors across the N-1 participants (training data, labeled data). The winner-take-all (WTA) classifier classifies the test vector to the category of the training vector that yields the highest correlation with the training vector (Fig 5A). For a given test pattern, correct classification yields a score of 1 and an incorrect classification yields a score of 0. For each left-out infant, we computed the percentage correct across all categories, and then the mean decoding performance across all participants in an age group (Fig 5B).”

**Reviewer #2 (Recommendations For The Authors):**
I only have some minor comments.Typo on line 90 ("Infants participants in 5 conditions, which [...]").

Thanks for pointing this out. We have now corrected ‘participants’ to ‘participated’.

Typo on lines 330: "[...] in example 4-5-months-olds.".

Thanks for pointing this out. We changed ‘4-5-months-olds’ to ‘4-5-month-olds’.

Figure 2 - bar plots: rotating and spacing out values on the x-axis may improve readability. Ditto for the line plots in Figure 4.

Thanks for the suggestions. In the revised manuscript, we have improved the readability of Figure 2.

Caption of Figure 6: description of the distinctiveness plots may refer to panel C, instead of the bottom panels of section B.

Thanks for pointing this out. We have now corrected this information in the manuscript.